# Sulfation affects apical extracellular matrix organization during development of the *Drosophila* embryonic salivary gland tube

**J Luke Woodward[1†‡], Jeffrey Matthew[1†], Rutuparna Joshi[1], Vishakha Vishwakarma[1§], Ying Xiao[2], SeYeon Chung[1]\***

[1]Department of Biological Sciences, Louisiana State University, Baton Rouge, United States; [2]Shared Instrumentation Facility, Louisiana State University, Baton Rouge, United States

**\*For correspondence:** seyeonchung@lsu.edu

[†]These authors contributed equally to this work

**Present address:** [‡]Department of Cell and Molecular Biology, University of Chicago, Chicago, United States; [§]Division of Developmental Biology, Eunice Kennedy Shriver National Institute of Child Health and Human Development, National Institute of Health, Bethesda, United States

**Competing interest:** The authors declare that no competing interests exist.

## eLife Assessment

This paper is **important** in demonstrating a requirement for sulfation in organizing apical extracellular matrix (aECM) during tubulogenesis in *Drosophila melanogaster*. The authors identify and characterize the organization of some of the first known components of the non-chitinous aECM in the *Drosophila* salivary gland tube, and these findings are supported by **convincing** data. This study would be of interest to developmental and cell biologists.
[Editors' note: this paper was reviewed by Review Commons.]

**Abstract** The apical extracellular matrix (aECM) plays a critical role in epithelial tube morphogenesis during organ formation, but its composition and organization remain poorly understood. Using the *Drosophila* embryonic salivary gland (SG) as a model, we identify Papss, an enzyme that synthesizes the universal sulfate donor PAPS, as a critical regulator of tube lumen expansion. *Papss* mutants show a disorganized apical membrane, condensed aECM, and disruptions in Golgi structures and intracellular trafficking. SG-specific expression of wild-type Papss, but not the catalytically inactive form, rescues the defects in *Papss* mutants, suggesting that defects in sulfation are the underlying cause of the phenotypes. Additionally, we identify two zona pellucida (ZP) domain proteins, Piopio (Pio), and Dumpy (Dpy), as key components of the SG aECM. In the absence of *Papss*, Pio is gradually lost in the aECM, while the Dpy-positive aECM structure is condensed and dissociates from the apical membrane, leading to a thin lumen. Mutations in *dpy* or *pio*, or in *Notopleural*, which encodes a matriptase that cleaves Pio to form the luminal Pio pool, result in a SG lumen with alternating bulges and constrictions, with the loss of *pio* leading to the loss of Dpy in the lumen. Our findings underscore the essential role of sulfation in organizing the aECM during tubular organ formation and highlight the mechanical support provided by ZP domain proteins in maintaining luminal diameter.

## Introduction

Epithelial tubes are critical components of many key biological organs, including the kidney, heart, and lungs. Aberrant tube size leads to common diseases and congenital disorders, such as polycystic kidney disease, asthma, and lung hypoplasia. Tube morphogenesis is regulated by multiple factors, including intracellular mechanisms, intercellular interactions, and interactions between cells and the

surrounding extracellular matrix (ECM). Specifically, events at the apical (luminal) surface of a tube play an important role in regulating tube diameter and growth, thereby influencing overall tube shape (*Myat and Andrew, 2002*). During tubular organ formation, the surrounding epithelium secretes aECM components into the lumen. These aECM components form the luminal meshwork and interact with the apical membrane, playing a critical role in epithelial morphogenesis and regulation of tube dimensions (*Jaźwińska et al., 2003*; *Devine et al., 2005*; *Luschnig et al., 2006*; *Wang et al., 2006*; *Syed et al., 2012*; *Dong et al., 2014*; *Drees et al., 2023*; *Gill et al., 2016*; *Cohen et al., 2020b*).

The aECM has a distinct composition compared to the basal ECM, with a diverse array of components (*Zheng et al., 2020*). ZP domain proteins, which are highly conserved across species, are of particular interest and are characterized by a conserved 260 amino acid ZP domain that mediates homo- or heteromeric polymerization into filaments. Initially identified in the extracellular coat of mammalian oocytes, ZP domain proteins have been shown to play a critical role in the organization of specialized apical structures in various tissues in species ranging from worms to humans (*Porter and Tamm, 1955*; *Yan et al., 2012*; *Legan et al., 1997*; *Beeck et al., 2012*; *Renner et al., 2007*; *McAllister et al., 1994*; *Wong et al., 2000*; *Cohen et al., 2020a*; *Plaza et al., 2010*; *Jovine et al., 2005*; *Litscher and Wassarman, 2020*; *Wilkin et al., 2000*; *Ghosh and Treisman, 2024*). Chitin, a biopolymer of N-acetylglucosamine, is another major aECM component that is abundant in invertebrates and some vertebrates (*Tang et al., 2015*; *Mei et al., 2024*). In many *Drosophila* and *C. elegans* tissues, chitin forms a network of fibers that provides mechanical supports and maintains the integrity of the epithelial layer (*Araújo et al., 2005*; *Devine et al., 2005*; *Moussian et al., 2006*; *Tonning et al., 2005*; *Zhang et al., 2005*). Several studies have shown the role of ZP proteins in maintaining the chitin matrix in the aECM. During *Drosophila* tracheal development, two ZP proteins, Dpy and Pio, build an elastic structure in the lumen that modulates mechanical stress and maintains the chitin matrix (*Jaźwińska et al., 2003*; *Dong et al., 2014*; *Drees et al., 2023*). In the *Drosophila* corneal lens, another ZP domain protein, Dusky-like, establishes an external scaffold for chitin retention (*Ghosh and Treisman, 2024*). However, many organs in both vertebrates and invertebrates have a non-chitinous aECM, and the role of ZP proteins in the non-chitinous aECM is less well understood.

Post-translational modifications of the aECM are essential for proper tube dimensions during organ formation (*Luschnig et al., 2006*; *Wang et al., 2006*; *Abrams et al., 2006*; *Herman and Horvitz, 1999*; *Hwang et al., 2003*). Sulfation is an abundant post-translational modification that takes place in the Golgi apparatus, where a sulfate group is covalently attached to amino acid residues of proteins or side chains of proteoglycans (*Günal et al., 2019*). Several studies in *C. elegans* have shown key roles of sulfation in neuronal organization, development of reproductive tissues, and phenotypic plasticity, linking this post-translational modification to multiple cellular processes (*Turnbull et al., 2003*; *Bülow and Hobert, 2004*; *Kinnunen et al., 2005*; *Tecle et al., 2013*; *Díaz-Balzac et al., 2014*; *Dierker et al., 2016*). A study in a mouse model of cystic fibrosis also suggests a link between the disease state and proper sulfation levels in organs (*Xia et al., 2005*), but the role of sulfation in tubular organ development and aECM organization remains elusive.

Sulfation reactions require a high-energy form of sulfate, 3'-phosphoadenosine-5'-phosphosulfate (PAPS), which is produced by the enzyme PAPS synthetase (Papss) (*van den Boom et al., 2012*; *Günal et al., 2019*). Mammals have two Papss isoforms, Papss1, and Papss2. *Papss1* shows ubiquitous expression, with higher levels in glandular cells and salivary duct cells, suggesting that high degrees of sulfation are required in these cell types (*Uhlén et al., 2005*; *Karlsson et al., 2021*). *Papss2* shows a more restricted expression in certain tissues, such as cartilage, and mutations in *Papss2* have been associated with skeletal dysplasia in humans (*Ahmad et al., 1998*; *Haque et al., 1998*; *Kurima et al., 1998*; *Ford-Hutchinson et al., 2005*). Mouse models with exon deletions in *Papss1* or *Papss2* show extensive defects, including lethality (*Papss1*) and abnormal craniofacial and limb morphology (*Papss2*) (*Groza et al., 2023*), demonstrating that Papss is vital for developmental processes in mammals. *Drosophila* has a single *Papss* gene, allowing for a straightforward investigation of its function in developmental processes without redundancy. By studying the role of *Papss* in *Drosophila* development, we can gain valuable insights into the role of this conserved enzyme in normal development and disease in humans.

To understand the role of post-translational modification in aECM organization and to determine the key composition of the non-chitinous aECM in tubular organs, we use the *Drosophila* embryonic SG as a model. This single-layered epithelial tube is formed through highly stereotyped

morphogenetic processes (*Chung et al., 2014*), making it an excellent system for elucidating the molecular mechanisms underlying aECM composition and modification. SG cells are specified on the ventral surface of the embryo at stage 10. Once specified, SG cells do not undergo cell division or cell death because SG-upregulated transcription factors downregulate the expression of key cell cycle genes and proapoptotic genes (*Myat and Andrew, 2000a*; *Chandrasekaran and Beckendorf, 2003*; *Fox et al., 2013*; *Matthew et al., 2024*), and thus the entire morphogenetic process is driven by cell shape change, cell rearrangement, and collective cell migration (*Myat and Andrew, 2000b*; *Bradley and Andrew, 2001*; *Bradley et al., 2003*; *Chung and Andrew, 2014*; *Chung et al., 2017*). At stage 11, SG cells invaginate to form a three-dimensional tubular structure, which then elongates and migrates toward its final position during stages 12–13. The SG continues to migrate, and the lumen expands significantly by apical secretion during stages 14–16, forming the fully developed organ. Despite limited knowledge of the composition of the aECM in the SG, previous work has demonstrated the critical role of ADAMTS-A, a secreted metalloprotease, in releasing cells from the aECM to facilitate SG migration and normal lumen shape (*Ismat et al., 2013*). Our previous work has also shown that cell-aECM adhesion via the apical surface protein Cad99C influences cell rearrangement during SG elongation (*Chung and Andrew, 2014*).

Here, we identify Papss as being essential for aECM organization and lumen expansion of the *Drosophila* embryonic SG. Loss of *Papss* results in a disorganized apical membrane and a condensed aECM, which is associated with disrupted Golgi structures and intracellular trafficking. *Papss* mutants also show defects in the basal ECM and occasional SG cell death, leading to invasion of hemocytes into SG epithelia that engulf dying cells. Expressing the wild-type (WT) Papss in the SG rescues all phenotypes, but not the enzyme-dead form. We identify two ZP proteins, Pio and Dpy, as key components of the SG aECM. In the SG lumen, these ZP domain proteins form a filamentous scaffold that is significantly affected by the loss of *Papss*, with Pio eventually being lost from the aECM, and the Dpy-positive aECM structure condensing and dissociating from the apical membrane. Mutations in *pio* or *dpy*, or in *Notopleural* (*Np*), which encodes a matriptase that cleaves Pio to form the luminal pool of Pio, exhibit bulging and constriction of the SG lumen. Notably, mutations in *pio* result in the loss of luminal Dpy signals in the SG. These data indicate that the Dpy-containing filamentous structure of the aECM is crucial for maintaining luminal diameter and that the luminal pool of Pio is essential for organizing this structure during SG morphogenesis.

## Results

### Papss is required for SG lumen expansion

To identify key enzymes that affect SG morphology by modifying aECM components, we examined the effects of the loss of ~20 enzymes that are highly expressed in the SG and are likely to function in the secretory pathway (*Fox et al., 2013*). Deficiency lines deleting each enzyme were immunostained with the apical membrane marker Crumbs (Crb) or Stranded at Second (SAS) to assess SG lumen shape. Among the twelve lines showing defects (*Supplementary file 1*), a deficiency line deleting *PAPS synthetase* (*Papss*) and 18 other genes had a very thin SG lumen with irregular apical protrusions (*Figure 1A*). Papss catalyzes the synthesis of 3'-phosphoadenosine-5'-phosphosulfate (PAPS), the sulfate donor compound in all sulfotransferase reactions (*Venkatachalam, 2003*). Consistent with a previous report (*Jullien et al., 1997*), our fluorescence in situ hybridization confirmed that *Papss* transcripts are highly expressed in the SG throughout morphogenesis (*Figure 1B*), suggesting a high demand for sulfation in this organ.

To test the role of *Papss* and confirm the phenotype, we used a *Papss* EMS mutant line (*Papss²*; *Zhu et al., 2005*). The Papss protein contains two critical protein domains, the ATP sulfurylase domain, which adds sulfate to an AMP group of ATP to form adenosine 5'-phosphosulfate (APS), and the APS kinase domain, which phosphorylates APS to form PAPS (*Venkatachalam et al., 1998*; *Mueller and Shafqat, 2013*). Sanger sequencing of this mutant identified a point mutation (G->A) that introduces a premature stop codon (Trp271->stop; amino acid number is for the Papss-PA isoform) located near the N-terminus of the ATP sulfurylase domain, resulting in a truncated protein in which most of the domain is not translated (*Figure 1—figure supplement 1S1A-B*). Since both protein domains are essential for the enzyme function of Papss, we conclude that this mutant line is potentially null or a strong hypomorphic allele. *Papss* homozygous mutants and transheterozygotes for *Papss* mutant over

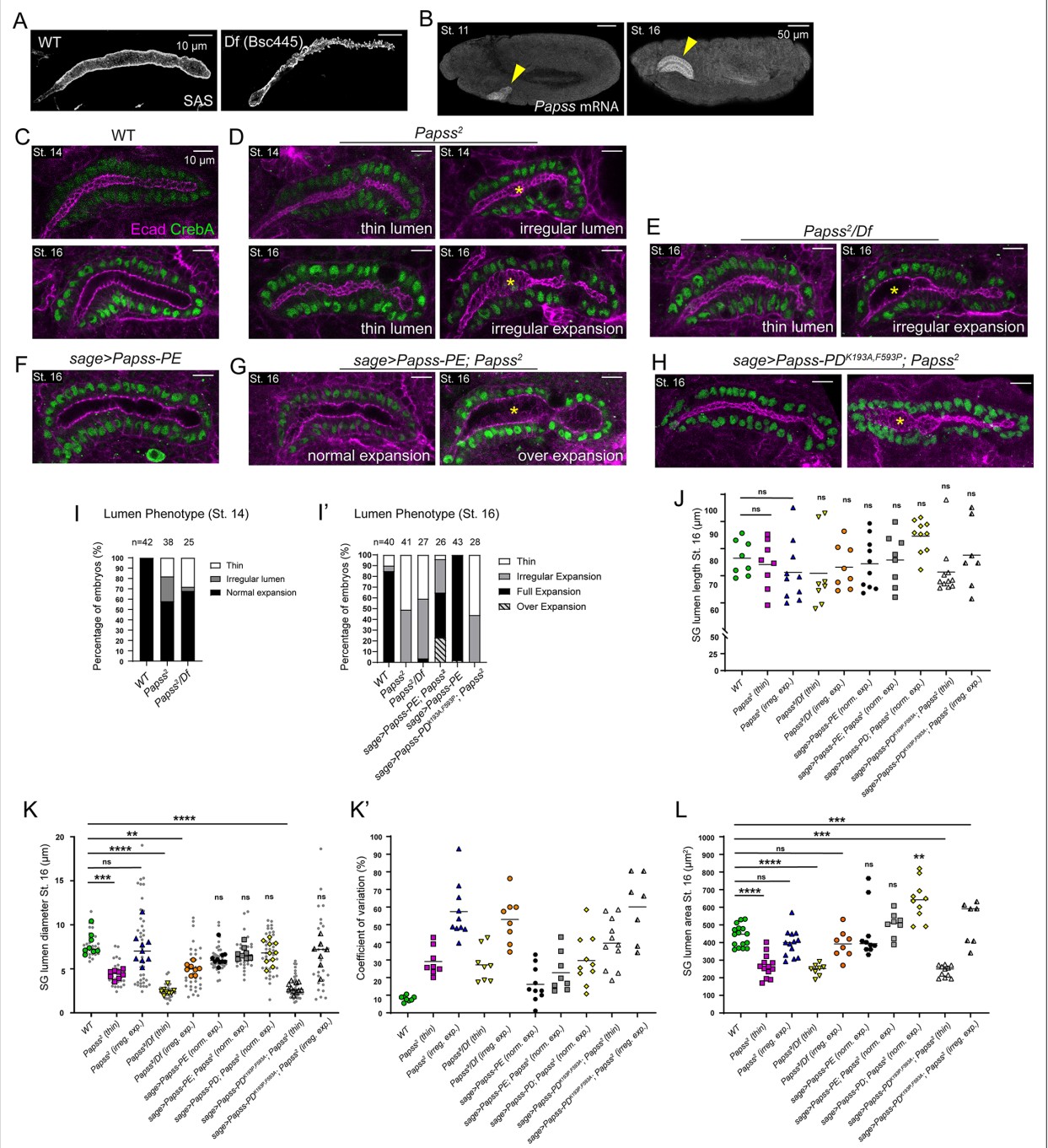

**Figure 1.** PAPS synthetase (Papss) mutants show defects in salivary gland (SG) lumen expansion. (**A**) Stage 16 SGs immunostained for Stranded at Second (SAS). (**B**) In situ hybridization using an antisense probe for *Papss* mRNA in wild-type (WT) embryos. (**C–H**) Stage 14 and stage 16 SGs immunostained for Ecad and CrebA. Yellow asterisks indicate regions of irregular, non-uniform expansion (**D, E, H**) or over-expansion (**G**) of the SG lumen. (**I, I'**) Quantification of different SG lumen phenotypes. (**J**) Quantification of SG lumen length. n=7–11 SGs for each genotype. See the Materials and methods for the sample size of each genotype. One-way ANOVA test with multiple comparisons (*p<0.05; ns, non-significant). (**K, K'**) Quantification of the SG lumen diameter (**K**) and the coefficient of variation for each SG (**K'**). The diameter is measured in three different regions of the lumen. The average diameter, as well as the individual data points, are shown. The same samples are used for the quantifications in J-K'. One-way ANOVA test with multiple comparison (***p<0.001; ns, non-significant) (**L**) Quantification of SG lumen area. n=7–16 SGs for each genotype. One-way ANOVA test with multiple comparison (***p<0.001; ****p<0.0001; ns, non-significant). Horizontal lines indicate mean values.

The online version of this article includes the following source data and figure supplement(s) for figure 1:

**Source data 1.** Raw data for the quantification of salivary gland (SG) lumen length, lumen diameter, lumen area, and coefficient variations for lumen

*Figure 1 continued*

diameter.

See *Figure 1J-L* for the graphs.

**Source data 2.** Raw data for the quantification of salivary gland (SG) lumen length, lumen diameter, lumen area, and coefficient variations for lumen diameter.

**Figure supplement 1.** *PAPS synthetase (Papss) mutants encode a stop codon in the ATP sulfurylase domain.*

**Figure supplement 2.** Sulfotyrosine signals are reduced in PAPS synthetase (*Papss)* mutants.

**Figure supplement 3.** Some PAPS synthetase (*Papss)* mutant embryos show defective salivary gland (SG) formation and whole embryonic defects.

**Figure supplement 3—source data 1.** Raw data for quantifying defects in the whole embryonic morphology and salivary gland (SG) positioning in wild-type (WT), Papss homozygous, transheterozygous over a deficiency, and heterozygous mutant embryos.

**Figure supplement 4.** Several splice forms of PAPS synthetase (*Papss*) are expressed in the salivary gland (SG).

a deficiency were embryonic lethal. To test the role of Papss in sulfation in the SG, we checked for the presence of sulfated tyrosine using a sulfotyrosine antibody in WT and *Papss* mutant embryos. In WT, we observed strong sulfotyrosine signals both in the apical membrane and as puncta in the cytoplasm of SG cells (*Figure 1—figure supplement 1S2A*). The cytoplasmic puncta colocalized with ManII-GFP signals, a medial Golgi marker (*Yang et al., 2021*), suggesting that proteins undergoing tyrosine sulfation in the Golgi are detected by the antibody (*Figure 1—figure supplement 1S2C*). In *Papss* mutants, the cytoplasmic punctate signals were significantly reduced (*Figure 1—figure supplement 2B*), suggesting that sulfation in the Golgi is decreased. The signals at the apical membrane appeared similar or slightly reduced compared to WT, suggesting that sulfated molecules, possibly by maternally provided Papss, were still targeted apically, although it is not clear whether they were properly targeted to their original destination or mislocalized. Together with a previous study using Alcian Blue staining showing the loss of sulfated biomolecules in the *Papss* mutant SG (*Zhu et al., 2005*), these data suggest sulfation in the SG is dependent on Papss.

*Papss* homozygous mutant embryos often showed overall defective embryonic morphology, such as defects in germ band retraction and head involution (39%; N=80/204; *Figure 1—figure supplement 3A–F*). The heterozygous *Papss* mutant embryos produced by crossing *Papss* mutants with WT flies appeared normal (*Figure 1—figure supplement 3A–F*). A smaller percentage of transheterozygous embryos showed similar defects (10%; N=13/136), with many late-stage embryos appearing normal (*Figure 1—figure supplement 3C, E and F*). We conclude that transheterozygous embryos with more severe defects did not survive to late stages, but we cannot rule out the possibility that a secondary mutation at another gene locus contributes to the overall embryonic defects. Embryos with these morphological defects often showed defects in SG invagination, forming a SG close to the surface of the embryo (*Figure 1—figure supplement 3D, G and H*). We used the embryos that formed the SG in the normal position for quantification and further analysis.

Using the adherens junction (AJ) marker E-cadherin (Ecad) signals, we analyzed SG lumen phenotypes in *Papss* homozygous mutant embryos and transheterozygotes. During stages 14–16, the WT SG lumen expanded consistently throughout its length (*Figure 1C*). In contrast, *Papss* mutant embryos exhibited irregularities in SG lumen expansion. At stage 14, when the WT SG lumen expansion began, more than 40% of *Papss* mutant embryos had either a thinner lumen (*Figure 1D and I*; 18%; N=7/38) or a slightly irregular lumen with bulges (*Figure 1D and I*; 24%; N=9/38). At stage 16, about half of *Papss* homozygous or mutant embryos or transheterozygotes over a deficiency showed an overall thin lumen with an irregular apical surface (*Figure 1D and I'*), and the other half had an irregularly expanded lumen with thin and expanded regions along its length (*Figure 1D and I'*). Quantification of the SG lumen length and diameter at stage 16 showed that whereas the lumen length was comparable between WT and *Papss* mutant SGs (*Figure 1J*), the diameter was significantly decreased or irregular, with a high degree of variation in *Papss* mutant SGs (*Figure 1K and K'*). Quantification of the SG lumen area at stage 16 revealed that *Papss* mutants' SGs exhibiting the thin lumen phenotype had a smaller lumen size compared to WT. Together, these data suggest a role for Papss in SG lumen expansion primarily by expanding the lumen diameter (*Figure 1L*).

To determine whether the SG lumen expansion defects are due to the loss of *Papss*, we performed a rescue experiment by overexpressing Papss in the SG using the SG-specific driver *sage-Gal4* (*Chung et al., 2009*) in the *Papss* mutant background. *Papss* produces multiple isoforms, including

Papss-PA/B/C/G (629 aa), Papss-PD (657 aa), and Papss-PE/F/H (630 aa), which differ only slightly in the N-terminus (Flybase; https://flybase.org/). Using three additional probes that detect different subsets of splice forms, we performed fluorescent in situ hybridization and found that all probes detected signals in the SG at varying levels (*Figure 1—figure supplement 4A and B*). We generated UAS lines for Papss-PD and Papss-PE and found that overexpression of either isoform in the SG of WT embryos resulted in a normally expanded SG lumen (data for Papss-PE overexpression are shown in *Figure 1F and I'*). Given the high levels of *Papss* mRNA in the WT SG, the Papss protein is likely abundant in the SG, which may explain why further overexpression of Papss does not lead to any additional effects. Since our initial analysis showed identical rescue features in the SG for the generated UAS lines, we used the UAS-Papss-PD and UAS-Papss-PE lines interchangeably for further experiments (hereafter, when overexpressed, Papss-PD and Papss-PE will be referred to as Papss). When Papss was overexpressed in the SG of *Papss* mutant embryos, only a small fraction (4%; N=1/26) showed a thin lumen phenotype along the entire length of the SG (*Figure 1G and I'*). The majority of embryos (73%) showed either a normal, fully expanded lumen along the entire length of the SG, with occasional slight irregularities at the apical membrane (42%; N=11/26), or an irregularly expanded lumen, where some parts of the SG lumen were expanded and others were thin (31%; N=8/26) (*Figure 1G and I'*). SGs with normally expanded lumens had luminal geometry comparable to that of WT embryos (*Figure 1J–L*). Interestingly, some of the embryos had an over-expanded SG lumen, often with a non-uniform diameter (24%; N=6/24; *Figure 1G and I'*). To test whether the enzyme activity of Papss is required for its role in SG lumen expansion, we also generated UAS lines with mutations in the critical residues of the ATP sulfurylase (F593P) and APS kinase (K193A) domains using the Papss-PD isoform (See Materials and Methods for details; amino acid number is for Papss-PD). Overexpressing a mutant version of Papss containing both of these mutations in the SG did not rescue the SG phenotypes as WT Papss did, but it showed the same range of defects as *Papss* mutants (*Figure 1H, I' and J-L*). These data suggest that the sulfation activity of Papss is required for SG lumen expansion. Overall, although a range of phenotypes was present, the thin SG lumen phenotype was almost completely rescued by SG-specific expression of WT Papss, but not the enzyme-dead form.

## Loss of *Papss* disrupts the apical domain and aECM organization

To investigate the cellular basis of the thin SG lumen phenotype in *Papss* mutants, we examined the localization of key apical membrane proteins and junctional markers. In WT SGs, the apical polarity protein Crb localized to the subapical region (SAR) between the apical surface (free apical area) and the AJs, and we observed clear Crb signals along the apical cell boundary of each SG cell (*Figure 2A and B*). In contrast, the Crb signals in *Papss* mutants appeared less prominently localized to the SAR in SG cells (*Figure 2C*). We measured the Crb signal intensity in the SAR and the apical-medial region of SG cells. The ratio of the Crb signal intensity at SAR to apical-medial intensity was significantly lower in *Papss* mutants than in WT (*Figure 2D*), indicating disorganization of the apical domain. The AJ marker Ecad and the septate junction (SJ) marker Discs large (Dlg) showed no apparent disruption in localization compared to WT (*Figure 2—figure supplement 1A–E*). These data suggest that loss of *Papss* primarily affects the apical membrane in the SG, while leaving the junctional regions intact.

Next, we investigated the organization of the aECM, using wheat germ agglutinin (WGA), which marks several glycan modifications, including O-linked β-N-acetylglucosamine (O-GlcNAc). WGA has been shown to label the apical membrane and luminal structure in the *Drosophila* embryonic trachea (*Tonning et al., 2005*; *Chung et al., 2009*). Similarly, we observed relatively weak WGA signals in the apical membrane and the lumen of WT SGs (*Figure 2E*; n=25 SGs). Notably, the luminal signals showed multiple thin filamentous organizations traversing the lumen (*Figure 2E*), suggesting heterogeneity of the luminal structure. In *Papss* mutants, WGA signals were increased in both the lumen and apical membrane, with the apical membrane signals appearing more diffuse (*Figure 2F*; n=28 SGs). Interestingly, the luminal WGA signals in *Papss* mutants accumulated strongly on one side along the entire length of the lumen, whereas the thin filamentous WGA signals in WT SGs appeared to show no bias and were mostly centered in the lumen. In regions with a thin SG lumen, WGA was present in a highly condensed rod-like structure; in expanded regions, WGA signals showed either a similar condensed rod-like structure, favoring one side of the SG lumen, or a slightly condensed filamentous structure often broadly associated with the apical membrane (*Figure 2F*). Transmission electron microscopy (TEM) analysis also revealed highly condensed and more electron-dense luminal

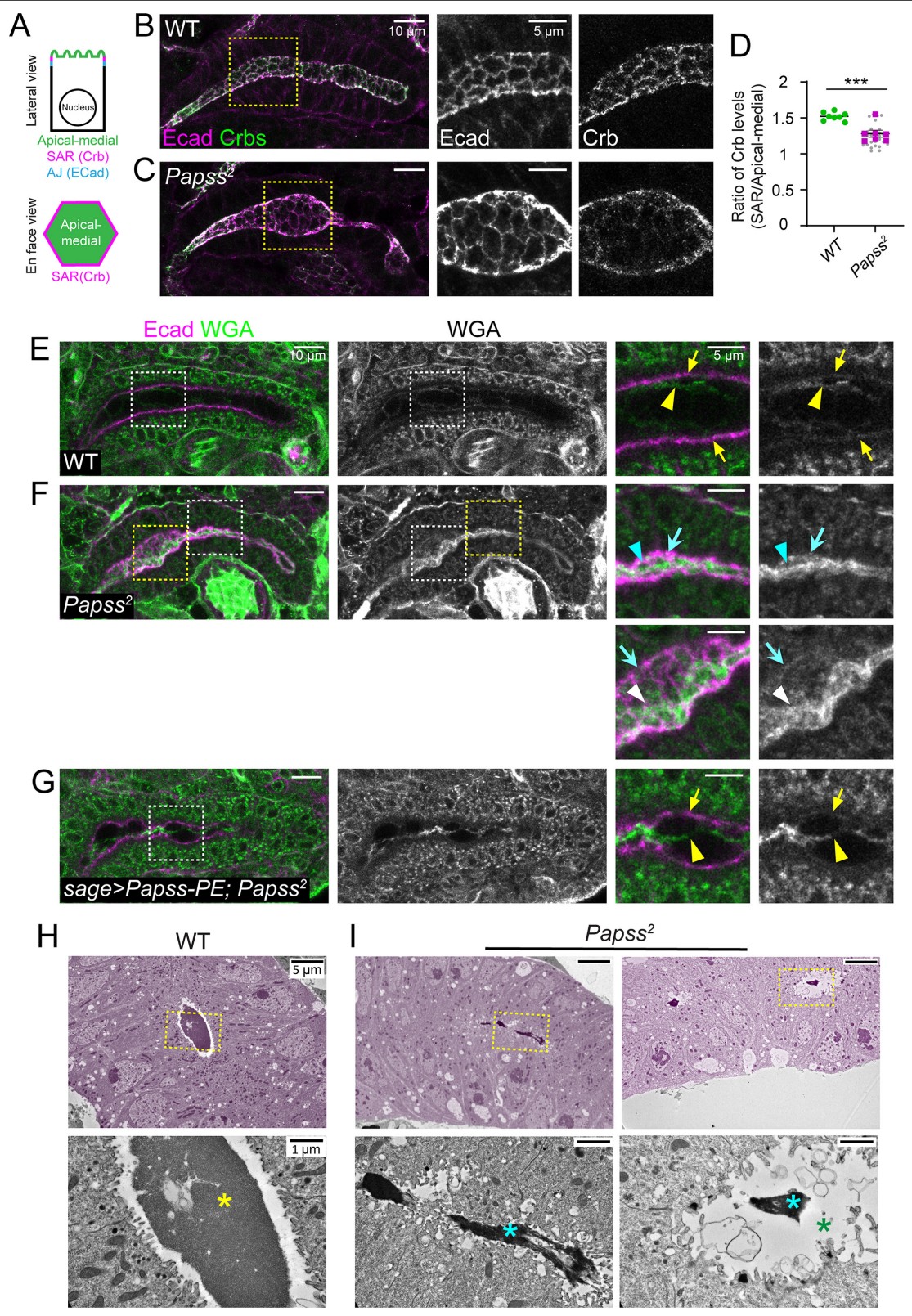

**Figure 2.** Mutations in PAPS synthetase (*Papss*) result in mislocalized Crumbs (Crb) and disruption of the apical extracellular matrix (aECM) architecture. (**A**) Cartoon showing the subcellular localization of Crb in salivary gland (SG) cells. (**B, C**) Stage 16 SGs immunostained for Crb and E-cadherin (Ecad). Magnified images are shown for the yellow dotted boxed regions. (**D**) Quantification of the Crb signal intensity ratio between the subapical region (SAR) and apical-medial regions of SG cells. (n=7, WT; n=6, *Papss²* (4–8 cells per SG)). Student's t-test with Welch's correction (\*\*\*p<0.001).

*Figure 2 continued on next page*

*Figure 2 continued*

(**E–G**) Stage 16 SGs stained for Ecad and wheat germ agglutinin (WGA). Magnified images are shown for the boxed regions. Arrows, WGA localization at the apical membrane in WT (yellow) and *Papss* mutant (cyan) SGs. Yellow arrowhead in E, WGA localizes as thin filaments in the lumen of WT SG. Cyan arrowheads, WGA localizes as a highly condensed structure in the thin luminal region of the *Papss* SG. White arrowheads, WGA localizes as a condensed, filamentous structure in the expanded luminal region of the *Papss* SG. Yellow arrowheads in G, WGA localizes as a mildly condensed filament in the lumen of *sage >Papss PE; Papss²* SG. (**H, I**) Transmission electron microscopy (TEM) images of stage 15 SGs. SGs are pseudo-colored in magenta. Magnified images of the luminal areas (boxed regions) are shown at the bottom. Yellow asterisk in H, electron-dense luminal material in WT SG. Cyan asterisks in I, condensed electron-dense luminal material in the *Papss* mutant SG. Green asterisk, a large luminal area lacking electron-dense material in the *Papss* mutant SG.

The online version of this article includes the following source data and figure supplement(s) for figure 2:

**Source data 1.** Raw data for the quantification of the mean gray values of Crumbs (Crb) levels and the ratio of the values between the subapical and apical-medial regions.

**Figure supplement 1.** Adherens junctions (AJs) and septate junctions (SJs) are intact in PAPS synthetase (*Papss*) mutants.

**Figure supplement 2.** Original, raw transmission electron microscopy (TEM) images of wild-type (WT) and *Papss* mutant salivary glands (SGs).

structures in the SG of *Papss* mutants compared to WT (***Figure 2H and I***; original, raw TEM images of several WT and *Papss* mutant samples are shown in ***Figure 2—figure supplement 2A and B***). SG-specific overexpression of Papss in the *Papss* mutant background resulted in a partial rescue; the luminal WGA structures were less condensed than *Papss* mutants, but they were not as thin as those in WT SGs (***Figure 2G***). Taken together, these data suggest a role for Papss in organizing the SG aECM, with loss of *Papss* resulting in a tightly packed or condensed aECM that is loosely associated with the apical membrane.

## Loss of *Papss* disrupts Golgi organization and mislocalizes intracellular trafficking markers

In addition to the apical membrane and luminal signals, we detected WGA signals around the nucleus, consistent with WGA binding to glycoproteins on the nuclear envelope (***Rutherford et al., 1997***), and in several large, punctate cytoplasmic structures in WT SG cells (***Figure 3A and E***). The cytoplasmic WGA puncta colocalized with the medial-Golgi marker ManII-GFP, suggesting that WGA binds glyco-proteins in the Golgi (***Figure 3J and L***). The number and intensity of the punctate WGA signals were significantly reduced in *Papss* mutants (***Figure 3B, F and I′***). Overexpression of WT Papss, but not the mutant form of Papss, in the SG restored the number of WGA cytoplasmic puncta (***Figure 3C, D, G, H and 'I'***). The intensity of the WGA signals was not restored in either case (***Figure 3I′***).

Interestingly, we observed highly dispersed ManII-GFP signals in *Papss* mutants SGs (***Figure 3K and L′***), suggesting that Golgi structures may be disrupted upon *Papss* loss. To examine the Golgi structures in more detail, we performed TEM analysis in WT and *Papss* mutant SG cells. In contrast to the mammalian Golgi, the *Drosophila* Golgi lacks the characteristic ribbon-like structure and is composed of a series of individual cisternae arranged in a random, disordered fashion (***Kondylis and Rabouille, 2009***). While the Golgi in WT SG cells showed randomly arranged cisternae, the Golgi in *Papss* mutants showed abnormal arrangement, either stacked or compressed, with small electron-dense puncta along these structures (***Figure 3M and N***), suggesting a functional defect and possible protein aggregation. Other organelles, such as the endoplasmic reticulum (ER), appeared relatively normal. SG-specific expression of GFP-KDEL, GFP fused to the ER retention motif KDEL, or immu-nostaining for prolyl-4-hydroxylase-alpha SG1, an ER-resident enzyme (***Abrams et al., 2006***), showed no significant differences between WT and *Papss* mutant SGs (***Figure 3—figure supplement 1A-D***). TEM analysis also revealed similar ER structures in WT and *Papss* mutant SG cells (***Figure 3—figure supplement 1E and F***). These data suggest that loss of *Papss* disrupts the Golgi, but not the ER.

To determine whether the phenotypes observed in *Papss* mutants are due to defects in Golgi struc-ture and function, we examined the morphology of the SG in null mutants of two key Golgi compo-nents, GM130 and Grasp65 (***Zhou et al., 2014***). Both mutants were homozygous-viable. Although the GM130 null allele (*GM130△²³*) was extremely weak and difficult to grow, the embryos we managed to collect appeared normal, and the SG exhibited only mild irregularity (***Figure 3—figure supplement 2A***; n=5/5). It is possible that only relatively normal embryos survived until late stages of embryogen-esis. However, *Grasp65* null mutants (*Grasp65¹⁰²*) exhibited a highly irregular SG lumen with condensed

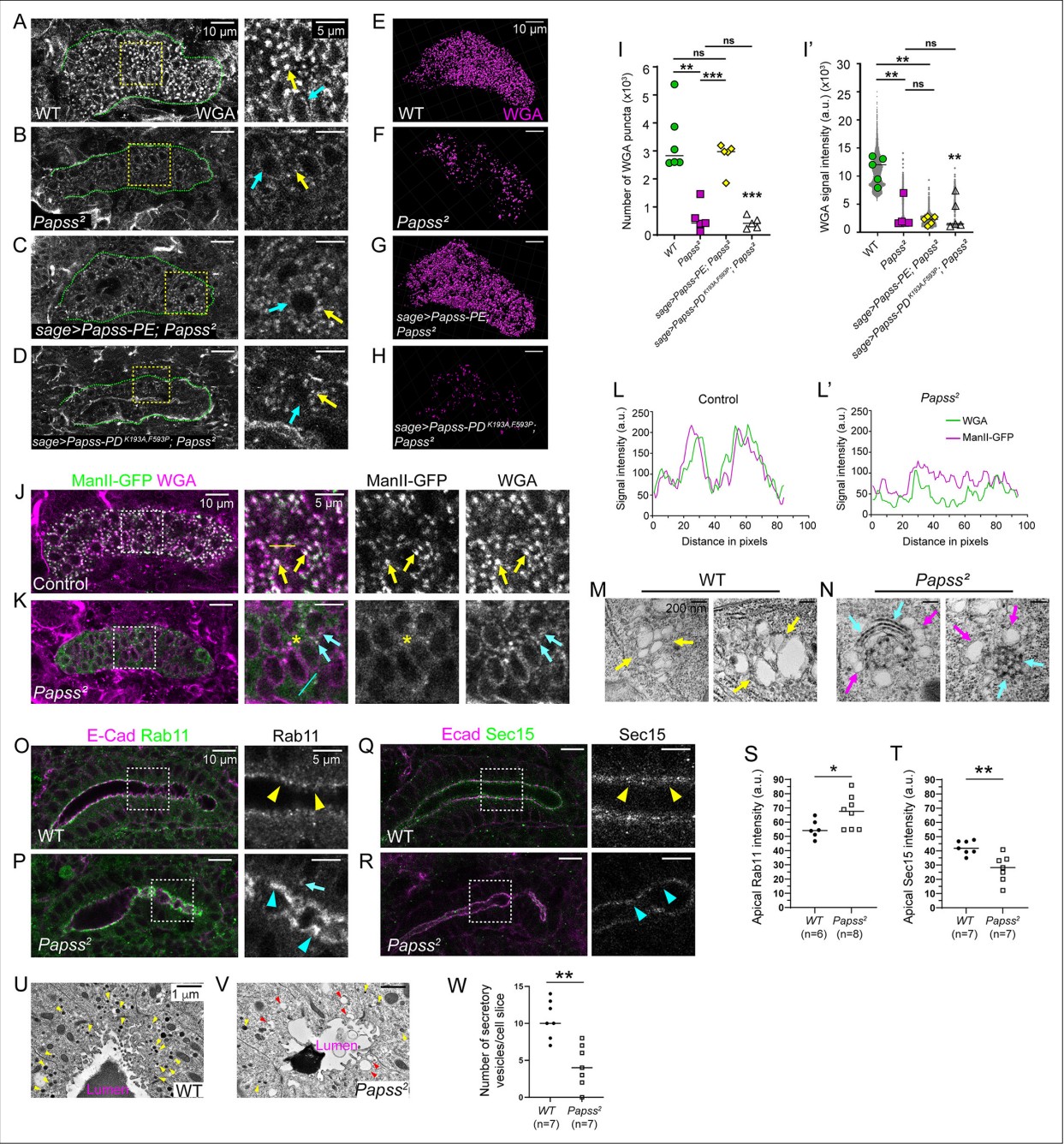

**Figure 3.** Loss of PAPS synthetase (*Papss*) results in disorganized Golgi structures and defects in intracellular trafficking components. (**A–D**) Stage 16 salivary glands (SGs) stained with wheat germ agglutini (WGA). Magnified images are shown for the boxed regions. Yellow arrows, WGA signals as cytoplasmic puncta. Cyan arrows, WGA signals in the nuclear envelope. (**E–H**) Three-dimensional (3D) reconstruction of cytoplasmic WGA puncta in stage 16 SGs. (**I, I'**) Quantification of the number of WGA puncta (**I**) and signal intensity (**I'**) within SG cells. n=5 SGs for all genotypes, except WT in G (n=6). One-way ANOVA test with multiple comparisons (**p<0.01; ***p<0.001; ns, non-significant). Horizontal lines indicate mean values. (**J, K**) Stage 16 SGs stained for GFP (for ManII-GFP) and WGA. Magnified images are shown for the boxed regions. Yellow arrows in J, ManII-GFP, and WGA puncta colocalize in WT SG. Asterisks in K indicate dispersed ManII-GFP signals in *Papss* mutants. Cyan arrows in K, little colocalization of ManII-GFP and WGA in the *Papss* SG. (**L, L'**) Line intensity profile showing the signal intensity of ManII-GFP (green) and WGA (magenta) along the yellow and cyan dotted lines in J and K, respectively. (**M, N**) Transmission electron microscopy (TEM) images of the Golgi. Yellow arrows in M, Golgi structures in wild-type (WT) SG cells. Magenta arrows in N, Golgi structures in the *Papss* SG. Cyan arrows in N, electron-dense structures in the Golgi of *Papss* mutant SG cells. (**O, P**) Stage 16 SGs immunostained for Rab11 and E-cadherin (Ecad). Magnified images are shown for boxed regions. Rab11 signals at the apical membrane are increased in *Papss* mutants (cyan arrowheads in P) compared to WT (yellow arrowheads in O), with occasional large punctate signals mislocalized to the basolateral domain (cyan arrow in P). (**Q, R**) Stage 16 SGs immunostained for Sec15 and Ecad. Compared to strong apical Sec15

*Figure 3 continued on next page*

*Figure 3 continued*

signals in the WT SG (yellow arrowheads in Q), Sec15 signals are reduced in *Papss* mutants (cyan arrowheads in R). (**S, T**) Quantification of the intensity of Rab11 (**S**) and Sec15 (**T**) at the apical membrane regions in WT and *Papss* mutant SGs. Welch's t-test (**p<0.05). Horizontal lines indicate mean values. (**U, V**) TEM images for the apical region of SG cells. Yellow arrowheads, electron-dense vesicles in both genotypes. Red arrowheads, empty, larger vesicle-like structures in *Papss* mutants. (**W**) Quantification of the electron-dense vesicle numbers per cell slice. Welch's t-test (**p<0.05). Horizontal lines indicate mean values.

The online version of this article includes the following source data and figure supplement(s) for figure 3:

**Source data 1.** Raw data used to quantify the number and intensity of wheat germ agglutini (WGA).

**Source data 2.** Raw data used to test the colocalization of wheat germ agglutini (WGA) and ManII-GFP signals.

**Source data 3.** Raw data for the quantification of Rab11, Sec15, and Rab7 intensities, as well as the number of Rab7-positive puncta.

**Source data 4.** Raw data for the number of electron-dense secretory vesicles per transmission electron microscopy (TEM) slice are shown in *Figure 3W*.

**Figure supplement 1.** The endoplasmic reticulum (ER) is intact in PAPS synthetase (*Papss*) mutants.

**Figure supplement 2.** Mutants of the Golgi component Grasp65 show an irregular salivary gland (SG) lumen phenotype.

**Figure supplement 3.** Rab7 punctate number is decreased in PAPS synthetase (*Papss)* mutants.

luminal WGA signals (*Figure 3—figure supplement 2B and C* and C; n=10/10). These data suggest that disrupted Golgi structures in *Papss* mutants may contribute to the SG defects.

The aberrant Golgi structures observed in *Papss* mutants prompted us to investigate the impact on intracellular trafficking components. We examined the localization of Rab11, a marker for recycling endosomes, Rab7, a marker for late endosomes, and Sec15, an exocyst marker. As previously shown (*Chung and Andrew, 2014*; *Le and Chung, 2021*), Rab11 and Sec15 were enriched along the apical domain of WT SG cells (*Figure 3O and Q*). In *Papss* mutants SGs, we observed an increase in apical Rab11 signal intensity, with some signals mislocalized to the basolateral region (*Figure 3P and S*). In contrast, Sec15 signals were significantly reduced at the apical membrane (*Figure 3R and T*). Unlike Rab11 or Sec15, Rab7 localized in large punctate structures in the cytoplasm of WT SG cells (*Figure 3—figure supplement 3A*). In *Papss* mutants, the number of Rab7 puncta was significantly reduced, although the intensity of Rab7 puncta was not significantly different from WT (*Figure 3—figure supplement 3B–D*). TEM analysis revealed a number of electron-dense vesicles in the apical domain of SG cells in WT (*Figure 3U*). In contrast, *Papss* mutant SGs showed fewer electron-dense vesicles, but instead, we observed several larger, empty vesicle-like structures (*Figure 3V and W*). Collectively, these data suggest that defective sulfation affects Golgi structures and multiple routes of intracellular trafficking.

## Loss of *Papss* results in abnormal cell death and disruption of the basal ECM

In *Papss* mutant embryos, we often observed one or more acellular regions within the SG epithelium during late embryogenesis that lacked any markers for nuclei, cell membranes, or junctions. To test whether these acellular regions were caused by cell death, we stained embryos with an antibody against *Drosophila* cleaved death caspase-1 (DCP-1; *Sarkissian et al., 2014*). Indeed, DCP-1 signals were often found within the acellular spaces (*Figure 4A and B*), suggesting that loss of *Papss* leads to abnormal cell death in the SG. TEM analysis also captured a dying cell in the *Papss* mutant SG, supporting the occurrence of cell death (*Figure 4C*). We found that *Papss* mutant SGs have a significantly higher number of these acellular spaces compared to WT SGs (*Figure 4D*). Quantifying the number of SG cells at stage 16 revealed that *Papss* homozygous embryos and transheterozygotes have significantly fewer cells than WT (*Figure 4E*). Overexpression of WT Papss, but not the mutant form of Papss, in the *Papss* mutant background decreased the number of acellular regions within the SG (*Figure 4D*) and restored the number of SG cells (*Figure 4E*). This suggests that loss of sulfation leads to conditions that hinder SG cell survival. Blocking apoptosis in the *Papss* mutant SG by overexpressing the baculoviral protein p35, a caspase inhibitor of apoptosis, also restored the SG cell number (*Figure 4E*), suggesting that apoptosis is a key mechanism underlying SG cell loss in *Papss* mutants. Importantly, the SGs in these embryos still showed thin (25%; N=2/8) or irregularly expanded (75%; N=6/8) lumens, suggesting that SG lumen expansion defects in *Papss* mutants are independent of abnormal cell death.

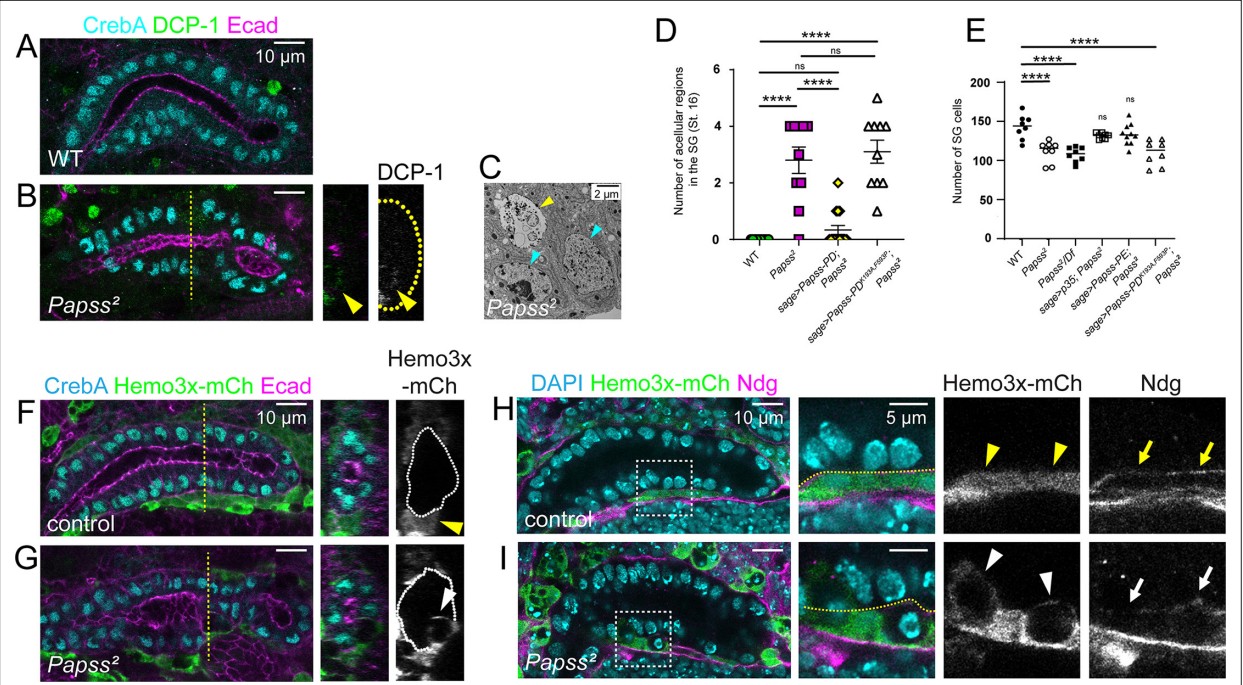

**Figure 4.** PAPS synthetase (Pappss) mutants show premature salivary gland (SG) cell death. (**A, B**) Stage 16 SGs immunostained for CrebA, DCP-1, and E-cadherin (Ecad). Cross-sectional images are shown for the yellow dashed line in B. Yellow arrowheads indicate that DCP-1 signal is detected in the SG in *Papss* mutants. White dashed lines, the basal boundary of the SG. (**C**) Transmission electron microscopy (TEM) image of stage 15 SG cells, including a dying cell (yellow arrow). Cyan arrows, nuclei of normal living cells. (**D**) Quantification of the number of acellular regions in stage 16 SGs. (n=8 SGs, WT; n=10, *Papss²*; n=15, *sage >Papss PD; Papss²*; n=8, *sage >Papss-PD^{K193A,F593P}; Papss²*) (**E**) Quantification of the number of SG cells at stage 16 (n=10 SGs, WT; n=10, *Papss²*; n=8, *Papss²/Df*; n=8, *sage >p35; Papss²*; n=10, *sage >Papss PE; Papss²*; n=8, *sage >Papss-PD^{K193A,F593P}; Papss²*). One-way ANOVA test (****p<0.0001; ns, non-significant). (**F, G**) Stage 16 SGs immunostained for CrebA, Ecad, and mCherry (for Hemo3x-mCh). Cross-section images are shown for the yellow dashed lines. White dashed lines, the basal boundary of the SG. Yellow arrowhead in F, hemocyte signals outside of the control SG. White arrowhead in G, hemocyte signals are present inside the SG in *Papss* mutants. (**H, I**) Stage 16 SGs immunostained for DAPI, mCh (for Hemo3x-mCh), and Ndg. Magnified views are shown for boxed regions. Yellow dashed lines, the basal boundary of the SG. (**H**) In WT, hemocytes are found outside of the SG (yellow arrowheads), with Ndg forming a barrier at the basal boundary of the SG (yellow arrows). (**I**) In *Papss* mutants, hemocytes invade the SG epithelia (white arrowheads), where Nidogen (Ndg) signals are irregular or absent (white arrows).

The online version of this article includes the following source data for figure 4:

**Source data 1.** Raw data for the number of acellular regions and of salivary gland (SG) cells shown in *Figure 4D and E*.

Using srpHemo3x-mCh (*Gyoergy et al., 2018*), we labeled hemocytes, which are analogous in function to macrophages in vertebrates. Hemocyte signals were often present in the acellular regions of the SG in *Papss* mutants, suggesting that hemocytes may aid in apoptotic cell clearance from the SG epithelium (*Figure 4F and G*). In normal epithelia, the basal ECM acts as a physical barrier, preventing hemocytes from migrating into the epithelial layer. To examine the integrity of the basal ECM in *Papss,* we used an antibody against Nidogen (Ndg), a glycoprotein with important linker functions in the basement membrane (*Wolfstetter and Holz, 2012*). Ndg-positive basal ECM was detected in a regular, thin layer around WT SGs at stage 16 (*Figure 4H*). However, in *Papss* mutants, Ndg signals were irregular and often disconnected, and hemocytes invaded SG cells through the gaps in the Ndg-positive basal ECM (*Figure 4I*). Overall, our data suggest that loss of *Papss* leads to aberrant cell death in the SG and disrupts both the apical and basal ECM of the SG.

## The ZP domain proteins Pio and Dpy of the SG aECM are affected by the loss of Papss

To better understand how aECM composition and organization are affected by *Papss* loss, we set out to identify SG aECM components. Since chitin, a major aECM component in many *Drosophila* tissues, is absent from the SG lumen (*Figure 5A*), we focused on ZP domain proteins. Of the 18 ZP domain proteins in *Drosophila*, at least six are expressed in the embryonic SG, including *pio, dpy, quasimodo*

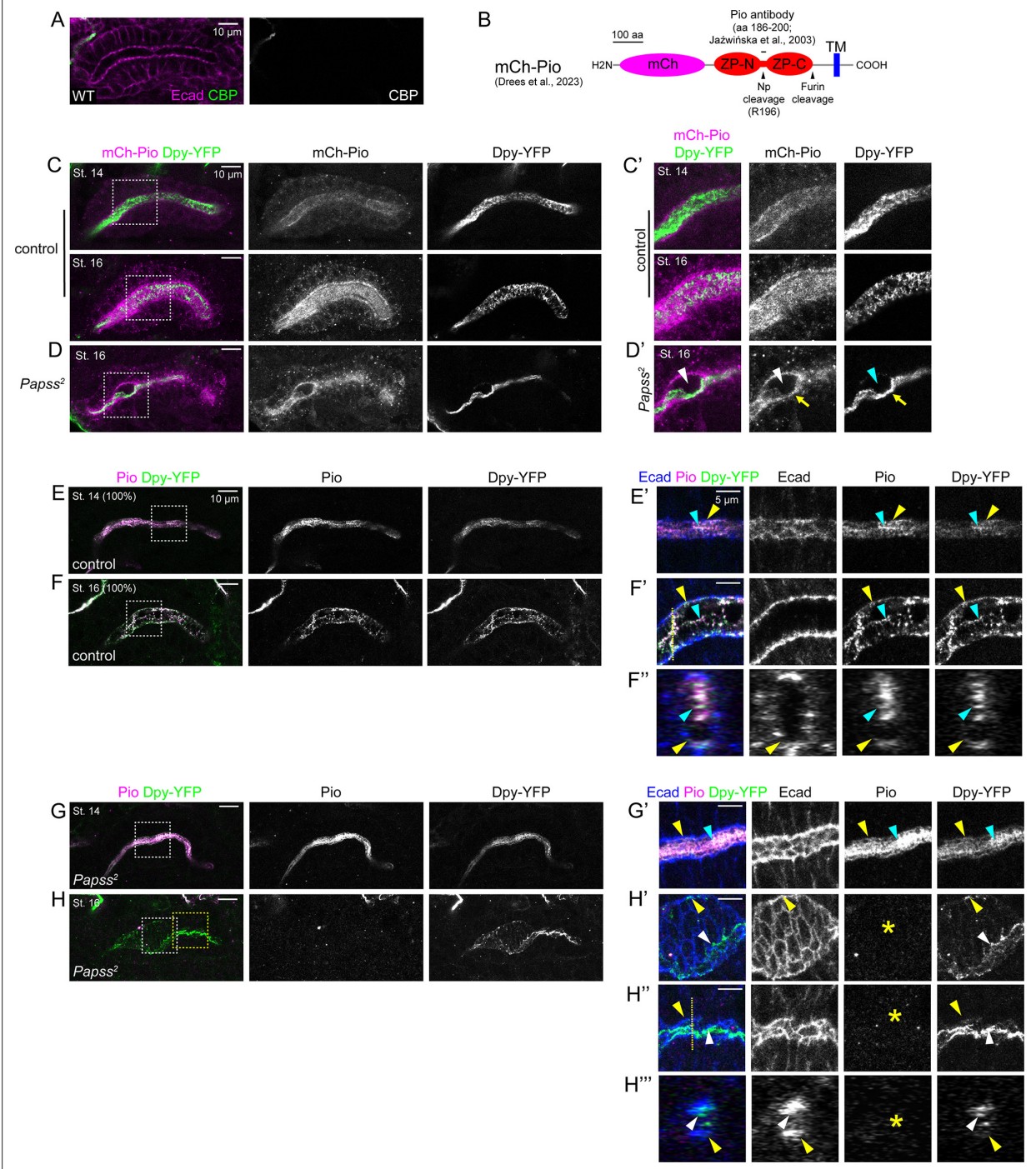

**Figure 5.** Loss of PAPS synthetase (*Papss*) disrupts the localization of Dumpy (Dpy) and Piopio (Pio) in the apical extracellular matrix (aECM). (**A**) Stage 16 salivary glands (SGs) stained for a chitin-binding protein (CBP) and E-cadherin (Ecad). (**B**) Cartoon of the mCh-Pio fusion protein showing the region for the antibody and protease cleavage sites. (**C, D**) Stage 14 and stage 16 SGs immunostained for mCh (for mCh-Pio) and GFP (for Dpy-YFP). The mCh-Pio (white arrowheads) and Dpy-YFP (cyan arrowheads) signals are largely absent from the expanded region of the lumen. Along one side of the lumen, weak mCh-Pio signals colocalize with condensed Dpy-YFP signals (yellow arrows). (**E-H'''**) Stage 14 and stage 16 SGs immunostained for Pio, GFP, and Ecad. Magnified images of boxed regions in E-H are shown in E'-H". F" and H''' are cross-sectional images of the SG lumen for regions indicated by white lines F' and H". Yellow arrowheads, Pio and Dpy localized to the apical membrane. Cyan arrowheads, Dpy and Pio localized to the SG lumen. Note a higher proportion of Dpy and Pio localized to the lumen in the *Papss* mutant SG at stage 14 (cyan arrowheads in G'). White arrowheads in H'-H''', Dpy accumulates towards one side of the lumen in the stage 16 *Papss* SG. Yellow asterisks, Pio is absent from the lumen at stage 16 in the *Papss* SG.

The online version of this article includes the following figure supplement(s) for figure 5:

*Figure 5 continued on next page*

*Figure 5 continued*

**Figure supplement 1.** Localization of Piopio (Pio) throughout salivary gland (SG) development.

**Figure supplement 2.** Dumpy (Dpy) localization through salivary gland (SG) morphogenesis.

**Figure supplement 3.** Quasimod (Qsm) localization patterns are not significantly affected in PAPS synthetase (*Papss*) mutants.

(*qsm*), *cypher* (*cyr*), *dusky* (*dy*), and CG17111 (**Tomancak et al., 2002**; **Jaźwińska and Affolter, 2004**; **Weiszmann et al., 2009**). We assessed the localization of Pio and Dpy, key aECM components critical for *Drosophila* tracheal morphogenesis (**Jaźwińska et al., 2003**; **Drees et al., 2023**), and Qsm, which plays a critical role in remodeling Dpy in the aECM during *Drosophila* flight muscle development (**Chu and Hayashi, 2021**).

To test Pio localization in the SG, we used a transgenic line expressing an mCherry-Pio (mCh-Pio) fusion protein with the N-terminal mCh tag (**Drees et al., 2023**; **Figure 5B**), as well as a Pio antibody generated with the peptide corresponding to the sequence within the ZP domain of Pio, between the ZP-N and ZP-C domains (**Jaźwińska et al., 2003**; **Figure 5B**). mCh-Pio signals were observed in cytoplasmic puncta, at the apical membrane, and in the SG lumen. These signals, especially the luminal signals, became stronger at stages 15–16 (**Figure 5C**), suggesting that Pio is transported apically and secreted into the SG lumen. Pio antibody signals showed distinct localization, predominantly in the SG lumen and at the apical membrane (**Figure 5E and F**). At stages 11 and 12, the Pio antibody revealed low levels of cytoplasmic punctate signals along with strong luminal signals, but from stage 13, Pio signals were predominantly localized to the SG lumen (**Figure 5E and F**; **Figure 5—figure supplement 1A**). Interestingly, compared to the diffuse luminal mCh-Pio signals (**Figure 5C**), Pio antibody signals in the SG lumen were heterogeneous with filamentous structures (**Figures 5E and 6F**), resembling luminal WGA signals. Pio antibody signals were absent at all stages in the *pio* null mutant, in which the entire ZP domain is deleted (**Drees et al., 2023**; **Figure 5—figure supplement 1B and C**), confirming antibody specificity. The Pio protein is known to be cleaved, at one cleavage site after the ZP domain, possibly by a furin protease, and at another cleavage site within the ZP domain by the matriptase Notopleural (Np) (**Drees et al., 2019**; **Drees et al., 2023**; **Figure 5B**). The latter cleavage site falls within the peptide sequence used to generate the Pio antibody (**Figure 5B**). A possible explanation for the different signals between mCh-Pio and Pio antibody signals is that the Pio antibody preferentially recognizes the C-terminal ZP domain fragment after furin and Np cleavage, whereas diffuse, luminal mCh-Pio signals are either the secreted Pio protein cleaved only by furin or the N-terminal fragment after Np cleavage.

Dpy-YFP signals from the Dpy-YFP knock-in line (**Lye et al., 2014**; YFP insertion into the *dpy* locus near the middle of the protein) showed heterogeneous, filamentous structures in the SG lumen throughout SG morphogenesis (**Figure 5—figure supplement 2A**) that colocalized with Pio antibody and WGA signals (**Figure 5E–F″**; **Figure 5—figure supplement 2C**). It should be noted that because homozygous Dpy-YFP embryos showed a bulging and constricted SG lumen phenotype with highly condensed Dpy-YFP signals throughout the lumen (**Figure 5—figure supplement 2B**), we used embryos with one copy of the transgene to analyze Dpy localization. Qsm localization was analyzed using the mCh-Qsm knock-in line, in which mCh is inserted into the *qsm* locus a few amino acids downstream of the N-terminal end of the protein (**Chu and Hayashi, 2021**). Unlike Pio and Dpy, mCh-Qsm signals were not detected in the SG lumen. Instead, they showed a dynamic intracellular localization ranging from cytoplasmic signals with apical membrane enrichment (stages 11–14) and apical vesicular signals (stages 14–15) to prominent signals at the apical membrane and SJs (stage 16) (**Figure 5—figure supplement 3A**). These data suggest that Qsm is not a component of the SG aECM. However, it is possible that the mCh-Qsm signals only detect the cleaved N-terminal fragment of Qsm, while the C-terminal fragment may localize to a different subcellular location. Taken together, our data suggest that Pio and Dpy form filamentous aECM structures in the SG.

We next examined Pio, mCh-Pio, Dpy-YFP, and mCh-Qsm signals in *Papss* mutant SGs at stages 14 and 16. At stage 14, Pio antibody and Dpy-YFP signals appeared condensed and enriched within the SG lumen, with reduced association with the apical membrane compared to WT (**Figure 5G and G′**). Interestingly, Pio antibody signals were barely detectable in the SG lumen at stage 16 (**Figure 5H–H‴**), likely due to either protein degradation or epitope masking caused by aECM condensation and protein misfolding. In contrast, Dpy-YFP signals remained in the lumen at all stages and were condensed and

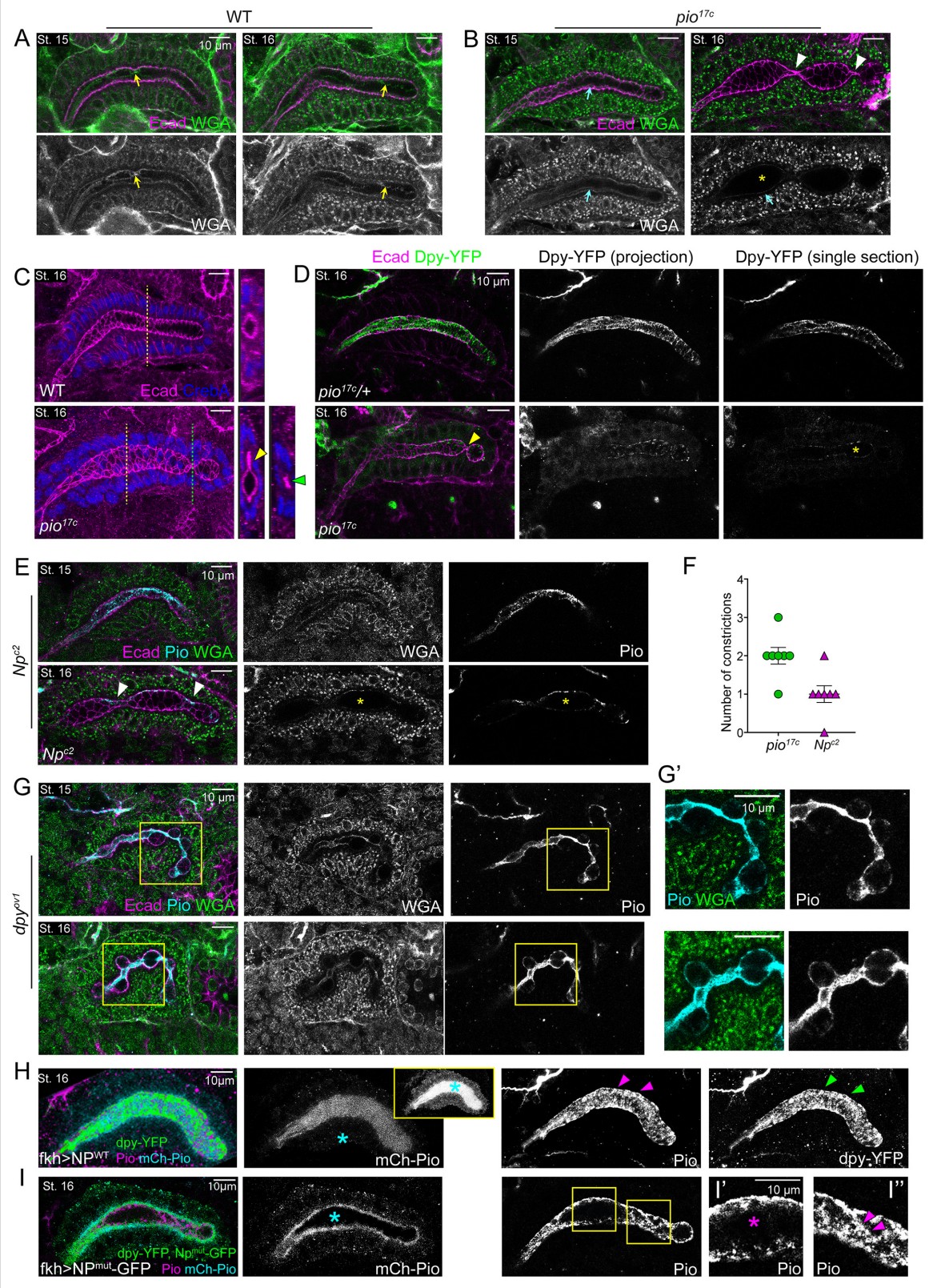

**Figure 6.** Piopio (Pio) is essential for a uniform salivary gland (SG) lumen diameter by maintaining Dumpy (Dpy) in the lumen. (**A, B**) SGs at stages 13–16 stained for wheat germ agglutini (WGA) and E-cadherin (Ecad). Yellow arrowheads in A indicate filamentous luminal WGA signals. Cyan arrowheads in B, WGA signals at the apical membrane. Yellow asterisks indicate loss of lumina WGA signals. White arrowheads, constricted lumen. (**C**) Stage 16 SGs immunostained for Ecad and CrebA. Cross-sectional images for the yellow and green dotted lines are shown on the right. Yellow arrowhead, a

*Figure 6 continued on next page*

*Figure 6 continued*

flat protrusion of the lumen. Green arrowhead, constricted lumen. (**D**) Stage 16 SGs stained for Ecad and GFP (for Dpy-YFP). Projection images contain merged z-sections, from the apical surface of the SG cells to midway through the SG. Single section, one z-section at the approximate midpoint of the lumen width. Yellow arrowhead, area of constricted lumen. Yellow asterisk, luminal area with no Dpy-YFP signals. (**E**) Single z-sections of confocal images of SGs at stages 15 and 16 stained for Ecad, Pio, and WGA. White arrowheads, area of constricted lumen. Yellow asterisks, no luminal signals of WGA and Pio. (**F**) Quantification of the number of constrictions in the SG lumen at stage 16 (n=7 for both genotypes). (**G, G'**) SGs at stages 14–16 stained for Ecad, WGA, and Pio. Higher magnification of the yellow boxed regions is shown in G'. (**H, I**) Stage 16 SGs overexpressing Np$^{WT}$ (**H**) or Np$^{mut}$ (**I**) that are immunostained for GFP (for Dpy-YFP and Np-GFP), Pio, and mCh (for mCh-Pio). Np$^{WT}$-overexpressing SGs show strong, uniform mCh-Pio signals in the lumen (cyan asterisks in H), whereas Np$^{mut}$-overexpressing SGs show no detectable signals in the lumen (cyan asterisk in I). In Np$^{WT}$-overexpressing SGs, the luminal mCh-Pio signals are so bright that the cytoplasmic, punctate mCh-Pio signals are rarely detectable alongside the luminal signals. Increasing the brightness to visualize the cytoplasmic punctate signals shows very strong luminal mCh-Pio signals (inset in H). Np$^{WT}$ overexpression shows high levels of Pio (magenta arrowheads in H) and dpy-YFP (green arrowheads in H) signals in both the apical membrane and lumen. In contrast, Np$^{mut}$ overexpression shows strong Pio signals in the apical membrane and various levels of Pio in the lumen. (**I', I''**) Higher magnification of the yellow boxed regions. The lumen diameter of the region with undetectable luminal Pio levels is slightly larger (magenta asterisk) compared to the region with clear luminal Pio signals (magenta arrowheads).

The online version of this article includes the following source data and figure supplement(s) for figure 6:

**Source data 1.** Raw data for the number of constrictions in the salivary gland (SG) of Piopio (*pio*) and Notopleura (*Np*) mutants.

**Figure supplement 1.** Loss of Piopio (*pio*) affects Dumpy (Dpy) localization patterns, and vice versa, in the salivary gland (SG).

localized to one side of the lumen at stage 16 (*Figure 5H–H'''*), similar to WGA signals (*Figure 2E*). mCh-Pio signals were depleted from the expanded region of the lumen, but weak signals colocalized with condensed Dpy-YFP signals (*Figure 5D*). We did not observe any significant differences in the localization or intensity of mCh-Qsm signals in *Papss* mutants (*Figure 5—figure supplement 3B and C*). These data suggest that the luminal pool of Pio and Dpy, which form filamentous structures, is significantly affected by the loss of *Papss*, resulting in the loss of Pio and the condensation and dissociation of the Dpy-containing aECM structure from the apical membrane.

## ZP domain matrix is required for uniform SG lumen diameter

We next investigated the role of Pio and Dpy in SG morphogenesis by analyzing SG lumen morphology in *pio* and *dpy* mutants. Immunostaining with Ecad revealed that *pio* null mutants formed relatively normal SGs up to stage 15, but stage 16 SGs showed dramatically abnormal luminal morphology characterized by multiple constrictions, with a slight expansion of the lumen between constrictions (*Figure 6A, B and F*; *Figure 6—figure supplement 1B*). The constricted regions were severely narrowed to the point of almost no luminal space, and some luminal regions also showed a flat protrusion on one side of the SG lumen (*Figure 6C*). Notably, these abnormal luminal shapes were associated with a progressive loss of luminal WGA and Dpy-YFP signals (*Figure 6B and D*). In *pio* mutants, WGA signals were comparable to WT at stages 13 and 14 (*Figure 6—figure supplement 1A–B*), but appeared slightly reduced and diffuse in the SG lumen at late stage 14, and this reduction became more pronounced at stage 15 (*Figure 6A and B*). Dpy-YFP signals were more severely affected in *pio* mutants. Unlike the strong, filamentous luminal signals in the control (*Figure 5—figure supplement 1A*), Dpy-YFP signals were reduced and diffuse in the SG lumen and were detected in the cytoplasm of *pio* mutant SG cells from stage 12 to stage 14, with a conspicuous absence in the lumen at stage 15 (*Figure 6—figure supplement 1C*). By stage 16, both WGA and Dpy-YFP signals were completely absent from the lumen in *pio* mutants, whereas the signals were still present at the apical membrane and in the cytoplasm (*Figure 6B and D*). We conclude that WGA primarily detects Dpy in the SG lumen and that Pio is required for Dpy secretion to the SG lumen.

To test a role for Pio cleavage and localization in shaping the SG lumen, we analyzed embryos mutant for the matriptase Np, which cleaves Pio. In *Np* mutants, Pio and WGA signals were observed in both the apical membrane and lumen up to stage 14, similar to WT embryos (*Figure 6E*). However, at stage 16, Pio signals were observed closely associated with the apical membrane, with no luminal Pio signals present (*Figure 6E*), suggesting that Np is required for this luminal pool of Pio. Importantly, *Np* mutants exhibited the same bulging and constricting SG luminal phenotype at stage 16 as *pio* mutants (*Figure 6E and F*), indicating that the filamentous luminal Pio structure plays a crucial role in maintaining a uniform SG diameter. *Dpy* mutants (*dpy$^{ov1}$*; a spontaneous mutant with a transposable element insertion into intron 11) exhibited a more severe bulging and constriction phenotype in the

SG lumen compared to *pio* mutants. Until stage 14, SGs in *dpy* mutants often showed a curved shape, but otherwise had a relatively normal morphology and uniform diameter (*Figure 6—figure supplement 1D*). At stages 15 and 16, SG lumen lost regularity and uniformity, often exhibiting multiple bulges and constrictions (*Figure 6G and G'*). In constricting regions, we observed condensed Pio and WGA signals in the lumen, whereas in bulging regions, weak Pio and WGA signals were detected along the apical boundaries (*Figure 6G and G'*). These data suggest a disrupted luminal scaffold that correlates with the bulging and constricting luminal morphology.

To further investigate the role of Np in cleaving Pio to organize the aECM during SG morphogenesis, we overexpressed WT (Np^WT) and mutant (Np^mut) forms of Np using the UAS-Np-Strep and UAS-Np.S990A-GFP lines (*Drees et al., 2019*). It should be noted that the endogenous Np remains when either form of Np is overexpressed in the SG. Overexpression of Np^WT increased the intensity of luminal mCh-Pio signals (*Figure 6H*), whereas overexpression of Np^mut resulted in the absence of the mCh-Pio signals in the lumen but strong mCh-Pio signals close to the apical membrane and in the cytoplasmic puncta (*Figure 6I*), supporting the role of Np in cleaving Pio to generate the luminal pool of Pio. In Np^WT-overexpressing SGs, the overall intensity of Pio and Dpy-YFP signals also appeared increased at both the apical membrane and in the lumen compared to WT (*Figure 6H*). These embryos did not show obvious changes in the SG morphology. In Np^mut-overexpressing SGs, Pio signals appeared to be increased at the apical membrane, and various levels of filamentous luminal signals were detected (*Figure 6I–I″*). Importantly, the regions of the lumen lacking luminal Pio signals was always slightly expanded compared to the region with filamentous luminal Pio signals, supporting the idea that the luminal Pio pool is important for a uniform lumen diameter (*Figure 6I and I″*; n=11/13). Dpy-YFP signals could not be analyzed properly with Np^mut overexpression because of the GFP tag in the Np^mut transgene. However, we detected two distinct pools of fluorescent signals in the GFP channel. The first pool was cytoplasmic punctate signals that colocalized with mCh-Pio, which we confirmed are Np^mut-GFP signals in a separate staining, and the second pool was thin filamentous luminal signals that overlapped with Pio antibody signals, which we suspect are Dpy-YFP signals (*Figure 6I*). These data suggest that overexpressed Np^mut failed to cleave mCh-Pio and Pio, leading to their accumulation at and near the apical membrane, but the endogenous Np still cleaved some pool, resulting in low levels of filamentous luminal Pio structures. Taken together, our data suggest that Dpy and Pio form a luminal scaffold necessary for a uniform SG tube diameter and that Pio cleavage by Np is essential for forming this scaffold.

## Discussion

Our study highlights the critical role of Papss, a key enzyme that produces the universal sulfate donor PAPS, in aECM organization and lumen expansion in the *Drosophila* embryonic SG. *Papss* mutants show defective SG lumen expansion, accompanied by disruption of the apical domain of SG cells and the aECM (*Figure 7*). The loss of *Papss* leads to defects in Golgi and intracellular trafficking, abnormal cell death in the SG, and defective basal ECM, emphasizing the importance of sulfation in multiple developmental processes. We also identify the ZP domain proteins Dpy and Pio as key components of the SG aECM, where Dpy is a scaffolding aECM structure that provides mechanical support, and Pio is required for proper organization of this structure (*Figure 7*).

### Sulfation is critical for aECM organization and SG lumen expansion

During *Drosophila* SG morphogenesis, secreted cargos are modified in the Golgi, secreted apically, and deposited in the lumen to form the aECM meshwork. Our data show that sulfation is essential for the proper organization of the aECM and the formation of a normal SG luminal shape. Consistent with a high demand for sulfation in the SG, transcripts for *Papss* and several other key enzymes involved in sulfation reactions are also upregulated in the SG, including *slalom* (*sll*), which encodes the *Drosophila* PAPS transporter (*Kamiyama et al., 2003*; *Lüders et al., 2003*), and *pipe* (*pip*) and *tyrosylprotein sulfotransferase* (*Tpst*), which encode sulfotransferases. Notably, Alcian Blue-stained materials are absent from the SG in *Papss*, *sll*, or *pip* mutants (*Zhu et al., 2005*), suggesting that sulfation in the SG depends entirely on these enzymes. Our data show that the irregularly expanded SG lumen phenotype in *Papss* mutants is associated with the condensed aECM structures and reduced association of the aECM with the apical membrane (*Figures 1 and 2*). Inappropriate charge balance due to defective

sulfation of proteoglycans and glycoproteins, as well as protein misfolding or failure to bind to appropriate partners, may contribute to aggregation of aECM components and condensation of the aECM structure. The ubiquitously expressed human *Papss1* gene shows upregulation in several glandular organs and the salivary duct (*Uhlén et al., 2005*; *Karlsson et al., 2021*), suggesting a potentially conserved role for this enzyme in modifying secreted molecules and modifying aECM in these organs.

The defects in SG lumen expansion in *Papss* mutants are linked to defects in intracellular trafficking, characterized by aberrant Golgi structures and mislocalized or reduced levels of several vesicle markers (*Figure 3*). This suggests that the loss of *Papss* has a broader impact on the intracellular trafficking system, beyond the sulfation of specific substrate molecules. Mutants of the key Golgi component Grasp65 show a mildly irregular SG lumen (*Figure 3—figure supplement 3*), supporting the important role of the Golgi in establishing proper lumen dimensions during organ formation. Consistent with the idea that the secretory pathway plays a key role in trafficking aECM components to the lumen, several mutants in intracellular trafficking genes, such as *CrebA*, *Sec61β*, and *Sar1*, which encode a major transcription factor that boosts secretory capacity, a Sec61 translocon component, and a COPII vesicle formation component, respectively, have been shown to have a slightly thinner SG lumen (*Fox et al., 2013*; *Fox and Andrew, 2015*). However, the defects in *Papss* mutants differ from those in general secretion mutants. While the latter exhibit a uniformly thin lumen, altered localization of several subcellular organelles, including mitochondria and the ER, and reduced Golgi signals (*Fox and Andrew, 2015*), *Papss* mutants show an irregularly expanded lumen and primarily affect the Golgi, while other organelles remain largely intact (*Figure 3*, *Figure 3—figure supplement 1*). Some of the abnormally modified targets appear to be aggregated in the defective Golgi in *Papss* mutants, but many are still transported apically and secreted into the lumen, resulting in a highly condensed luminal aggregate.

## The role of ZP domain proteins in non-chitinous aECM in the SG

We are beginning to understand the structural components and biological functions of the non-chitinous aECM during tubular organ formation. For example, in the non-chitinous aECM of the *C. elegans* vulva and excretory duct, distinct aECM components, including zona pellucida (ZP) domain proteins, chondroitin proteoglycans, and glycoproteins, work together to form the lumens of these tubular organs (*Gill et al., 2016*; *Cohen et al., 2020b*). Our work reveals that two ZP proteins, Dpy and Pio, play essential roles in organizing the aECM in the *Drosophila* SG, highlighting the important role of ZP proteins in forming the non-chitinous luminal matrix during tubular organ formation. Dpy, a gigantic protein (>2 MDa) with hundreds of epidermal growth factor (EGF) repeats interspersed with a Dpy module, forms long filamentous structures (*Wilkin et al., 2000*). With these filamentous structures, Dpy plays an essential role in the organization and attachment of apical structures in several tissues, including the thoracic cuticle (*Wilkin et al., 2000*), embryonic trachea (*Jaźwińska et al., 2003*), pupal wings (*Bökel et al., 2005*), genitalia (*Smith et al., 2020*), and muscle tendon cells (*Chu and Hayashi, 2021*). In the SG, aberrantly formed or lost luminal Dpy structures correlate with abnormal luminal morphology (*Figures 5 and 6*, *Figure 5—figure supplement 2*). We propose that the Dpy filamentous structure provides mechanical support in the SG aECM and that proper organization of these structures is critical for the formation of a uniform tube diameter. WGA signals always colocalize with Dpy-YFP signals in WT and *Papss* mutants, and both signals are absent from the lumen in *pio* mutants. WGA may primarily detect Dpy in the SG lumen by binding sugar residues that are attached to numerous EGF modules of the Dpy protein. A caveat to the analysis of Dpy using the Dpy-YFP line is that it is a non-functional fusion protein. An antibody against Dpy or a new functional fusion protein will be required to confirm Dpy localization. Even with this caveat, our data with the Pio antibody suggest that Dpy forms the filamentous luminal structure with Pio. Pio is required to secrete and maintain Dpy in the lumen of the SG (*Figure 6*). This is consistent with a role for Pio in Dpy secretion in the *Drosophila* trachea (*Drees et al., 2023*) and suggests a conserved role for Pio in multiple tissues. Since the ZP domains of Pio and Dpy have been shown to bind directly to each other in vitro (*Drees et al., 2023*), it is likely that the binding of Pio and Dpy is required for the formation of filamentous luminal structures. This binding may require cleavage of Pio by the matriptase Np, which generates the filamentous luminal pool of Pio (C-terminal ZP fragment of Pio). *Pio* and *Np* mutants show similar bulging and constricting SG lumen morphology (*Figure 6*), suggesting that the generation of the C-terminal ZP fragment of Pio by Np is critical for the formation of the filamentous luminal structure with Dpy.

Papss plays an essential role in the formation of the filamentous Dpy/Pio structure (*Figure 5*). It will be important to learn if the defects of the Pio-Dpy matrix in *Papss* mutants are a direct consequence of the loss of sulfation on these proteins or if it's an indirect outcome. The ZP domain can mediate homomeric or heteromeric polymerization into filaments, and its structure is stabilized by disulfide bonds formed between conserved Cys residues responsible for the polymerization of ZP modules (*Jovine et al., 2002*). Defective disulfide bond formation due to defective sulfation may destabilize Pio-Dpy interactions. Additionally, hundreds of EGF modules in the Dpy protein are highly susceptible to O-glycosylation. Sulfation of the glycan groups attached to Dpy may be critical for its ability to form a filamentous structure. In the absence of sulfation, the glycan groups on Dpy may not properly interact with the surrounding materials in the lumen, resulting in an aggregated and condensed structure. However, additional key substrate(s) of Papss, such as mucins, may exist in the SG aECM that play a key role in lumen expansion and shape. The striking enrichment of *Papss* expression in the SG compared to other tubular organs, such as the trachea, implies that the key substrate(s) of Papss are likely unique to or more highly enriched in the SG aECM than in the tracheal aECM.

It will be interesting to determine the role of other ZP proteins highly expressed in the SG. Qsm, for example, has been found in the same intracellular vesicles as Dpy in *Drosophila* tendon cells and has been shown to physically interact with Dpy through their ZP domains (*Chu and Hayashi, 2021*). Qsm has been shown to remodel Dpy filaments that link tendon cells to the pupal cuticle, increasing their tensile strength (*Chu and Hayashi, 2021*). In the SG, we find that Qsm is mainly localized to cytoplasmic vesicles and AJ/SJ regions rather than being present in the SG lumen like Pio and Dpy. It should be noted that, like Pio, Qsm has a furin cleavage site near the transmembrane domain and another putative protease cleavage site in the ZP domain. Since the mCh-Qsm line contains the mCh insertion near the N-terminal end of the protein, the mCh-Qsm signals are likely those of the cleaved N-terminal fragment after cleavage by these proteases and will need to be confirmed using a western blot. Further investigation will be needed to determine the role of Qsm in junctional regions of SG cells. Also, the roles of other SG-upregulated ZP domain proteins remain to be elucidated.

In conclusion, our analysis of the *Drosophila Papss* gene, a single ortholog of human Papss1 and Papss2, reveals its multiple roles during SG development. We expect that these findings will provide important insights into the function of these enzymes in normal development and disease in humans. Our findings on the key role of two ZP proteins, Pio and Dpy, as major components of the SG aECM also provide valuable information on the organization of the non-chitinous aECM during organ formation.

## Materials and methods
### Fly stocks and genetics
The fly stocks used in this study are listed in *Supplementary file 2*. All the crosses were performed at 25 °C.

### Generation of transgenic lines
UAS lines for the Papss-PD (657 aa) and Papss-PE/F/H (630 aa; hereafter referred to as Papss-PE) isoforms were generated. Papss-PD and Papss-PE differ only slightly in the N-terminus. Two UAS clones, UFO09742 (DGRC Stock 1643418; https://dgrc.bio.indiana.edu//stock/1643418; RRID:DGRC_1643418) and UFO09743 (DGRC Stock 1643419; https://dgrc.bio.indiana.edu//stock/1643419; RRID:DGRC_1643419), were obtained from the *Drosophila* Genomics Resource Center. UFO09742 was generated from the RE15281 cDNA clone encoding the open reading frame (ORF) for the Papss-PD isoform. UFO09743 was generated using the RE03925 cDNA clone, which encodes the same ORF for the Papss-PE isoform. Each clone contains a C-terminal Flag-HA tag. Purified DNA for the UAS clones (500 ng/μl) was sent to GenetiVision (Houston, TX) for injection. GenetiVision performed PhiC31-mediated insertions using the attP40 site.

UAS-Papss mutant lines for the APS kinase and ATP sulfurylase domains were generated by site-directed mutagenesis using the UAS-Papss-PD construct (UAS-Papss-PD$^{K193A}$, UAS-Papss-PD$^{F593P}$, and UAS-Papss-PD$^{K193A, F593P}$). K161 and F550 in human Papss (K193 and F593 residues of *Drosophila* Papss-PD) were identified as being critical for the catalytic efficiency of the APS kinase and ATP sulfurylase domains, respectively (*Zhang et al., 2022*; *Zhang et al., 2023*), and were mutated to alanine

and proline, respectively. The UAS-Papss-PD construct was amplified using the NEB Q5 master mix and the following primers.

### Primers used to make *UAS-Papss-PD*$^{K193A}$

Primer forward: 5'-CCGGGACGTGGCGGGTCTATACAAG-3'
Primer reverse: 5'-GTTTCGCACACATCCAG-3'

### Primers used to make *UAS-Papss-PD*$^{F593P}$

Primer forward: 5'-GATCCTGCCCCCGCGTGTTGCTG-3'
Primer reverse: 5'-TCCATGCTGTCTAGTC-3'

Following amplification, a kinase, ligase, and DpnI (KLD) reaction was performed, and the mutant construct was transformed into DH5α competent cells. To screen for colonies containing a correct mutation, plasmid DNAs were isolated, and the ORF of *Papss* was sequenced using the following primers.

### Primers for sequencing the *Papss* ORF

**Forward primer 1:** 5'-TGACCGAGCAAAAGCACCAT-3'
**Forward primer 2:** 5'-CGAGGTGGCCAAGTTGTTTG-3'
**Forward primer 3:** 5'-CCCGAGTTCTACTTTCAGCGT-3'
**Forward primer 4:** 5'-GCCGGTGCGAACTTCTACAT-3'
**Reverse primer:** 5'-ATACCAGGTACGCCTCCAGT-3'

To ensure no secondary mutations are present, the Papss ORF containing the correct mutations was ligated with the original backbone of the UAS-Papss construct. The construct containing both the K193A and F593P mutations was generated by combining DNA fragments containing each mutation, which were obtained by digesting the respective K193A and F593P constructs, and ligating them to a UAS backbone. Midi-prepped DNA samples were sent to GenetiVision (Houston, TX) for a PhiC31 injection.

### Immunostaining and confocal microscopy

*Drosophila* embryos were collected on a grape juice agar plate. All embryos were dechorionated by incubation in 50% bleach for 3 min at room temperature. After dechorionation, embryos were fixed in either saturated heptane or standard formaldehyde fixative (7 ml heptane, 5.6 ml ddH$_2$O, 700 µl 37% formaldehyde, 700 µl 10x PBS). Saturated heptane was prepared by mixing heptane: 37% formaldehyde 1:1 and shaking vigorously. For immunofluorescence staining, embryos were fixed in saturated heptane for 40 min at room temperature, devitellinized by vortexing in 80% ethanol, and then washed three times in absolute ethanol. For staining using the horseradish peroxidase (HRP) reaction for visualization, embryos were fixed in formaldehyde fixative and devitellinized by vortexing in 100% methanol. Embryos were then stained with primary and secondary antibodies in PBSTB (1X PBS, 0.2% bovine serum albumin, and 0.1% Triton X-100). All antibodies used are listed in *Supplementary file 3*. Embryos were then mounted in Aqua-Poly/Mount (Polysciences, Inc). Confocal images were taken using a Leica TCS SP8 confocal microscope with 63x, NA 1.4 and 40x, NA 1.3 objectives. HRP images were taken using a Leica DM2500 microscope using a 20 x, 0.8 NA objective.

### Fluorescence in-situ hybridization

In situ hybridization in WT embryos was performed as described in *Knirr et al., 1999*. Embryos were fixed in a standard formaldehyde fixation, followed by devitellinization by 100% methanol. To make the *Papss* mRNA probe that detects all splice forms, a 799 bp portion of the *Papss* ORF was amplified using the *UAS-Papss-PD* construct as a template. To make isoform-specific mRNA probes, unique sequence segments common to specific *Papss* isoforms were identified using Flybase sequences and used to design primers for PCR amplification, using WT genomic DNA as a template. The primers are listed below.

| Papss isoform | Primer sequence | Product size |
|---|---|---|
| All isoforms | Forward Primer: 5'-ACAAGGATGCGGGTCTGAAG-3'<br>Reverse Primer: 5'-GATTTGGTGACACTATAGCGCCTTAACTCGTTGGGAG-3' | 799 bp |
| ABCFG | Forward primer: 5'-TTCGCAGTTTTGTTTGGAAT-3'<br>Reverse primer: 5'-GATTTAGGTGACACTATAGTGCTCTCCGGAACTTCACTCAT-3' | 106 bp |
| EFH | Forward primer: 5'-TTCGCTTTCGACGTTCGTCG-3'<br>Reverse primer: 5'-GATTTAGGTGACACTATAGCTTCTTTTGATATGGGGATAAAA-3' | 228 bp |
| DH | Forward primer: 5'-TTCACCTGCCCAACGTTTCG-3'<br>Reverse primer: 5'-GATTTAGGTGACACTATAGCAAGCGCTGTTGTTATTGTTAA-3' | 106 bp |

*SP6 RNA polymerase promoter sequence underlined

Digoxigenin (DIG)-labeled RNA probe was generated by transcribing amplified and purified DNA using SP6 RNA polymerase and DIG RNA labeling mix (Sigma). Following hybridization, the RNA probe was detected using a sheep α-DIG antibody (Enzo; ENZ-ABS302-0001; 1:1000). The secondary antibody, α-sheep IgG conjugated with biotin (Invitrogen; 1:500), was visualized using a tyramide reaction with Cy3 (Perkin Elmer Life Sciences, NEL753001kt; 1:50). The embryos were mounted in Aqua-Poly/Mount. Confocal images were taken using a Leica TCS SP8 confocal microscope with 63x, NA 1.4 and 40x, NA 1.3 objectives.

## Genomic DNA sequencing of the Papss open reading frame region in *Papss²* mutant

*Papss²* homozygous mutant embryos were sorted for GFP signals in the balancer chromosome, and genomic DNA was extracted from them. The open reading frame of *Papss* was amplified using the isolated genomic DNA as a template, with the following primers.

Primer forward *Papss*: 5'-CTACAGGTGGCGACGAATGT-3'
Primer reverse *Papss*: 5'-GCGGCAGGTTCTGGTAGTAG-3'

Sanger sequencing of the amplified DNA segment of *Papss* was performed using the primers below.

Forward primer 1: 5'-ACAGGTGGCGACGAATGTG-3'
Forward primer 2: 5'-TGACCGAGCAAAAGCACCAT-3'
Forward primer 3: 5'-CGAGGTGGCCAAGTTGTTTG-3'
Forward primer 4: 5'-CCCGAGTTCTACTTTCAGCGT-3'
Forward primer 5: 5'-GCCGGTGCGAACTTCTACAT-3'
Reverse primer: 5'-GCGGCAGGTTCTGGTAGTAG-3'

## Quantification of SG cell number

Confocal images immunolabeled with CrebA for stage 16 SGs were used. 3D images for SGs were generated using Imaris (Andor), and SG nuclei were counted using cell ID numbers. Data were displayed using GraphPad Prism, and statistical significance was determined by one-way ANOVA.

## Quantification of SG lumen area

Confocal images immunolabeled with Ecad for stage 16 SGs were used. Z-stacks were merged to encompass the entirety of the SG lumen. Using Fiji (NIH), images were set to scale according to the image properties listed in the LasX files. The polygon selection tool was used to outline the entirety of the SG lumen, excluding the SG duct. SG lumen areas were determined for WT, *Papss* thin, and *Papss* irregularly expanded, respectively. Statistical significance was determined using a nonparametric t-test, as the data did not fit a Gaussian distribution. Sample numbers are: n=16, WT; n=13, *Papss²* with a thin lumen; n=11, *Papss²* with an irregularly expanded lumen; n=8, *Papss²/Df* with a thin lumen; n=8, *Papss²/Df* with an irregularly expanded lumen; n=10, *sage >Papss* with a normally expanded lumen; n=8, *sage >Papss PE; Papss²* with a normally expanded lumen; n=10, *sage >Papss PD; Papss²* with a normally expanded lumen; n=11, *sage >Papss-PD*[K193A,F593P]; *Papss²* with a thin lumen; n=7, *sage >Papss-PD*[K193A,F593P]; *Papss²* with an irregularly expanded lumen.

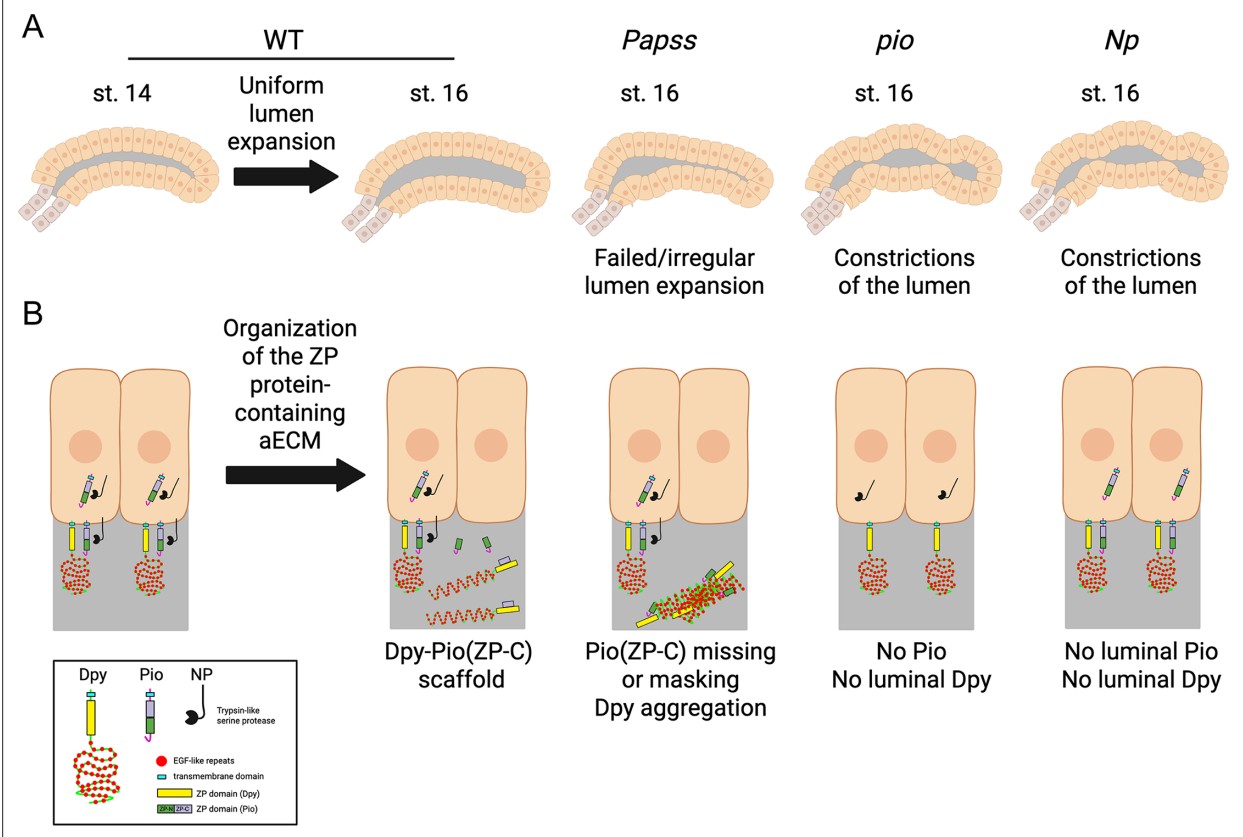

**Figure 7.** A proposed model. (**A**) Cartoon diagram showing normal lumen expansion in wild-type (WT) salivary gland (SG) and defective luminal morphology in PAPS synthetase (*Papss*), Piopio (*pio*), and Notopleura (*Np*) mutants. (**B**) Organization of the zona pellucida (ZP) domain protein-containing apical extracellular matrix (aECM) in the SG. In WT, the ZP-C fragment of Pio (which is cleaved by Np and furin proteases) forms a complex with Dumpy (Dpy) to build a filamentous scaffold within the lumen. In *Papss* mutants, this fragment is absent from the lumen at stage 16, and the Dpy-containing aECM structure becomes aggregated and highly condensed. The absence of luminal Pio in *pio* and *Np* mutants results in the loss of luminal Dpy and lumen constrictions. Created with BioRender.com.

## Quantification of SG lumen length and diameter

Confocal images of stage 16 SGs immunolabeled with an antibody for the cell junction marker, E-cad, were used. Several Z-sections, spanning approximately half of the circumferential/radial axis of the SG lumen, were merged to create a projection which covered the entire length of the lumen. The line tool in Fiji was used to measure the length of the SG lumen. To measure the diameter of the SG lumen, the lumen was divided into four equal segments using the lumen length obtained above, as there are sometimes minor variations in the width value along the length of the lumen. The line tool in Fiji was then used to measure the width along each quarter of the lumen, and an average value was obtained per SG. Each measurement obtained in Fiji was adjusted for scale according to the image properties in the raw confocal file. Sample numbers are: n=8, WT; n=8, *Papss²* with a thin lumen; n=10, *Papss²* with an irregularly expanded lumen; n=8, *Papss²/Df* with a thin lumen; n=8, *Papss²/Df* with an irregularly expanded lumen; n=10, *sage >Papss* PE with a normally expanded lumen; n=8, *sage >Papss PE*; *Papss²* with a normally expanded lumen; n=10 *sage >Papss PD*; *Papss²* with a normally expanded lumen; n=11, *sage >Papss-PD$^{K193A,F593P}$*; *Papss²* with a thin lumen; n=7, *sage >Papss-PD$^{K193A,F593P}$*; *Papss²* with an irregularly expanded lumen.

## Quantification of Crb signal intensity

Confocal images of stage 16 SGs immunolabeled with antibodies against the sub-apical region (SAR) marker, Crb, and the AJ marker, Ecad, were used. A maximum intensity projection of the apical region of several SG cells with the highest Ecad and Crb signals was created using seven z-sections. Regions were manually drawn along the inner or outer boundary of the Ecad signals of each cell to mark the

SAR and apical-medial regions. The mean gray value of the apical-medial Crb signals in each cell (the region delineated by the inner boundary of the Ecad signals) was quantified using Fiji. The mean gray value of the SAR Crb signals was calculated using the integrated densities and areas calculated with Fiji for regions between the inner and outer boundaries. To subtract the background, the average mean gray value of Crb signals for five regions outside the embryo was subtracted from the mean gray values of the SAR and apical-medial Crb signals in each SG cell. The ratio of the SAR to the apical-medial Crb level was compared. Seven and six embryos were used for the WT and *Papss* mutants, respectively, and four to eight cells per SG. SuperPlots were used to address variability in the data sets of each SG. Statistical significance was determined using a Welch's t-test.

### Quantification of the number and signal intensity of WGA puncta

High-resolution confocal images of stage 16 SGs immunostained with CrebA (SG nuclei), Ecad, and WGA were obtained using an Olympus SpinSR10 Sora spinning disk confocal microscope with a 100 x, NA 1.5 objective. Each image was taken under identical imaging conditions. Three-dimensional (3D) reconstruction of the WGA puncta was obtained using the Imaris software (Andor), and the number of puncta per SG was counted by Imaris. The Imaris machine learning tool was used to exclude non-puncta WGA signals within the SG and non-SG WGA puncta. After 3D reconstruction, the pixel intensity of each WGA punctum was also obtained and used for further analysis.

### Quantification of WGA and ManII-GFP signal intensity

Confocal images of stage 16 SGs immunostained for WGA and ManII-GFP were used. Single z-sections of each image were imported into Fiji, and a line intensity profile was generated using the Line Tool and Color Profiler plugin. Lines were drawn across regions spanning two ManII-GFP/WGA puncta, and the signal intensities for each pixel along the line for both ManII-GFP and WGA channels were obtained.

### Quantification for Rab11 and Sec15 apical region intensity

Single z-sections halfway through the proximal-distal axis from confocal images for stage 16 SGs immunolabeled for Rab11 or Sec15 and Ecad were used for quantification. Threshold images for Sec15 or Rab11, which display pixels as black or white based on whether they are above a value for signal intensity, were generated in Adobe Photoshop. In addition, Sec15 and Rab11 images were converted using the RGB stack function in Fiji. A stack of images, including the threshold and RGB stack converted images for Sec15 or Rab11 and the Ecad images, was generated using the images to stack function in Fiji. Using the threshold images for Sec15 and Rab11, a boundary for the apical region encompassing five SG cells, and the mean gray value intensity was calculated from the RGB stack converted Rab11 or Sec15 images. The graph was generated in GraphPad Prism, and statistical significance was calculated using Welch's t-test.

### Quantification for Rab7 puncta number and intensity

Single z-sections halfway through the proximal-distal axis from confocal images for stage 16 SGs immunolabeled for Rab7 and Ecad were used for quantification. Threshold images for Rab7, which display pixels as black or white based on whether they are above a value for signal intensity, were generated in Photoshop. Rab7 images were converted using the RGB stack function in Fiji. A stack of images, including the threshold and RGB stack converted images, for Rab7 and Ecad images were generated using the images to stack function in Fiji. Using the threshold images for Rab7, a boundary was drawn around all puncta present in five cells for each sample, and the mean gray value intensity and area was calculated for the RGB stack Rab7 converted image. The average mean gray value of punctate intensity and the average number of puncta in each cell were calculated using Fiji. Puncta with an area greater than or equal to $0.100 \mu m^2$ were used for quantification. Statistical significance was calculated by Welch's t-test using GraphPad Prism.

### Quantification of the number of dying cells

Confocal images of stage 16 SGs that were immunolabeled with antibodies against CrebA (to mark the SG nucleus), and DCP-1 (a cell death marker) were used. The number of regions within the SG

with DCP-1 signals and/or blank spots where CrebA signals should have been present were counted manually.

## Transmission electron microscopy (TEM)

Stage 15 embryos were fixed in 2% glutaraldehyde in 0.2 M phosphate buffer, pH 7.4 and rinsed four times in 0.1 M phosphate buffer, pH 7.2, containing 0.02 M glycine. Embryos were post-fixed in 2% osmium tetroxide for 1 hr in the dark, rinsed three times in dH$_2$O, stained en bloc in 1% uranyl acetate for 1 hr in the dark, and rinsed three times in dH$_2$O. Samples were dehydrated in a series of ethanol and then propylene oxide twice. Embryos were then infiltrated in propylene oxide: Epon resin series and embedded in Epon at 60 °C for 24 hr. Semi-thin sections were performed, followed by Toluidine Blue staining to obtain a salivary gland section. Ultra-thin sections for TEM were cut on a Leica EM UC7 microtome. TEM sections were mounted on carbon-coated copper grids (EMS FCF-150-CU), stained with 2% uranyl acetate and Reynolds lead citrate, and the ultra-thin sections were viewed on a JEOL JEM-1400, 120kV transmission electron microscope and imaged with a Gatan digital camera. Sections from at least five independent embryos were analyzed for wild-type and *Papss* mutants.

## Materials availability

All data and materials generated or analyzed during this study are available from the corresponding author.

## Acknowledgements

We thank the members of the Chung laboratory for their comments and suggestions. We thank M Behr, S Hayashi, the Bloomington stock center (NIH P40OD018537), and the Kyoto *Drosophila* stock center at Kyoto Institute of Technology for fly stocks. We thank M Affolter for the Pio antibody. Many antibodies were obtained from the Developmental Studies Hybridoma Bank, created by the NICHD of the NIH and maintained at the University of Iowa, Department of Biology, Iowa City, IA 52242. cDNAs for Papss were obtained from the Drosophila Genomics Resource Center (NIH 2P40OD010949). Confocal and electron microscopy were performed at the Shared Instrumentation Facility (SIF) at Louisiana State University. The model in *Figure 7* was created in https://BioRender.com. This work was supported by the National Science Foundation (MCB 2141387) and the National Institutes of Health (R03DE034704) to SC.

## Additional information

### Funding

| Funder | Grant reference number | Author |
| --- | --- | --- |
| National Science Foundation | MCB 2141387 | SeYeon Chung |
| National Institutes of Health | R03DE034704 | SeYeon Chung |

The funders had no role in study design, data collection and interpretation, or the decision to submit the work for publication.

### Author contributions

J Luke Woodward, Jeffrey Matthew, Formal analysis, Investigation, Writing – original draft, Writing – review and editing; Rutuparna Joshi, Formal analysis, Investigation; Vishakha Vishwakarma, Ying Xiao, Investigation; SeYeon Chung, Conceptualization, Formal analysis, Supervision, Funding acquisition, Investigation, Writing – original draft, Writing – review and editing

### Author ORCIDs

J Luke Woodward ⓘ https://orcid.org/0009-0006-2866-9533
Jeffrey Matthew ⓘ https://orcid.org/0000-0001-5686-6370
Rutuparna Joshi ⓘ https://orcid.org/0009-0007-4066-3748

Vishakha Vishwakarma http://orcid.org/0000-0001-6011-9845
SeYeon Chung https://orcid.org/0000-0002-5493-6424

Reviewer #1 (Public review): https://doi.org/10.7554/eLife.108292.3.sa1
Reviewer #2 (Public review): https://doi.org/10.7554/eLife.108292.3.sa2
Author response https://doi.org/10.7554/eLife.108292.3.sa3

---

## Additional files

### Supplementary files

Supplementary file 1. A targeted Df screen revealed Papss as a key enzyme for SG lumen expansion.

Supplementary file 2. Fly strains used.

Supplementary file 3. Antibodies used.

MDAR checklist

### Data availability

All data generated or analysed during this study are included in the manuscript and supporting files.

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
