## [Editor Report · eLife Assessment]

This paper is **important** in demonstrating a requirement for sulfation in organizing apical extracellular matrix (aECM) during tubulogenesis in *Drosophila melanogaster*. The authors identify and characterize the organization of some of the first known components of the non-chitinous aECM in the *Drosophila* salivary gland tube, and these findings are supported by **convincing** data. This study would be of interest to developmental and cell biologists.

[Editors' note: this paper was reviewed by Review Commons.]

---

## [Referee Report · Reviewer #1 (Public review)]

Summary:

There is growing appreciation for the important of luminal (apical) ECM in tube development, but such matrices are much less well understood than basal ECMs. Here the authors provide insights into the aECM that shapes the Drosophila salivary gland (SG) tube and the importance of PAPSS-dependent sulfation in its organization and function.

The first part of the paper focuses on careful phenotypic characterization of papss mutants, using multiple markers and TEM. This revealed reduced markers of sulfation and defects in both apical and basal ECM organization, Golgi (but not ER) morphology, number and localization of other endosomal compartments, plus increased cell death. The authors focus on the fact that papss mutants have an irregular SG lumen diameter, with both narrowed regions and bulged regions. They address the pleiotropy, showing that preventing the cell death and resultant gaps in the tube did not rescue the SG luminal shape defects and discussing similarities and differences between the papss mutant phenotype and those caused by more general trafficking defects. The analysis uses a papss nonsense mutant from an EMS screen - I appreciate the rigorous approach the authors took to analyze transheterozygotes (as well as homozygotes) plus rescued animals in order to rule out effects of linked mutations. Importantly, the rescue experiments also demonstrated that sulfation enzymatic activity is important.

The 2nd part of the paper focuses on the SG aECM, showing that Dpy and Pio ZP protein fusions localize abnormally in papss mutants and that these ZP mutants (and Np protease mutants) have similar SG lumen shaping defects to the papss mutants. A key conclusion is that SG lumen defects correlate with loss of a Pio+Dpy-dependent filamentous structure in the lumen. These data suggest that ZP protein misregulation could explain this part of the papss phenotype.

Overall, the text is very well written and clear. Figures are clearly labeled. The methods involve rigorous genetic approaches, microscopy, and quantifications/statistics and are documented appropriately. The findings are convincing.

Significance:

This study will be of interest to researchers studying developmental morphogenesis in general and specifically tube biology or the aECM. It should be particularly of interest to those studying sulfation or ZP proteins (which are broadly present in aECMs across organisms, including humans).

This study adds to the literature demonstrating the importance of luminal matrix in shaping tubular organs and greatly advances understanding of the luminal matrix in the Drosophila salivary gland, an important model of tubular organ development and one that has key matrix differences (such as no chitin) compared to other highly studied Drosophila tubes like the trachea.

The detailed description of the defects resulting from papss loss suggests that there are multiple different sulfated targets, with a subset specifically relevant to aECM biology. A limitation is that specific sulfated substrates are not identified here (e.g. are these the ZP proteins themselves or other matrix glycoproteins or lipids?); therefore, it's not clear how direct or indirect the effects of papss are on ZP proteins. However, this is clearly a direction for future work and does not detract from the excellent beginning made here.

---

## [Referee Report · Reviewer #2 (Public review)]

Summary

This study provides new insights into organ morphogenesis using the Drosophila salivary gland (SG) as a model. The authors identify a requirement for sulfation in regulating lumen expansion, which correlates with several effects at the cellular level, including regulation of intracellular trafficking and the organization of Golgi, the aECM and the apical membrane. In addition, the authors show that the ZP proteins Dumpy (Dpy) and Pio form an aECM regulating lumen expansion. Previous reports already pointed to a role for Papss in sulfation in SG and the presence of Dpy and Pio in the SG. Now this work extends these previous analyses and provides more detailed descriptions that may be relevant to the fields of morphogenesis and cell biology (with particular focus on ECM research and tubulogenesis). This study nicely presents valuable information regarding the requirements of sulfation and the aECM in SG development.

Strengths:

- The results supporting a role for sulfation in SG development are strong. In addition, the results supporting the involvement of Dpy and Pio in the aECM of the SG, their role in lumen expansion, and their interactions, are also strong.

- The authors have made an excellent job in revising and clarifying the many different issues raised by the reviewers, particularly with the addition of new experiments and quantifications. I consider that the manuscript has improved considerably.

- The authors generated a catalytically inactive Papss enzyme, which is not able to rescue the defects in Papss mutants, in contrast to wild type Papss. This result clearly indicates that the sulfation activity of Papss is required for SG development.

---

## [Author Response]

The following is the authors’ response to the original reviews

**Reviewer #1 (Public review):**
Summary:There is growing appreciation for the important of luminal (apical) ECM in tube development, but such matrices are much less well understood than basal ECMs. Here the authors provide insights into the aECM that shapes the Drosophila salivary gland (SG) tube and the importance of PAPSS-dependent sulfation in its organization and function.The first part of the paper focuses on careful phenotypic characterization of papss mutants, using multiple markers and TEM. This revealed reduced markers of sulfation and defects in both apical and basal ECM organization, Golgi (but not ER) morphology, number and localization of other endosomal compartments, plus increased cell death. The authors focus on the fact that papss mutants have an irregular SG lumen diameter, with both narrowed regions and bulged regions. They address the pleiotropy, showing that preventing the cell death and resultant gaps in the tube did not rescue the SG luminal shape defects and discussing similarities and differences between the papss mutant phenotype and those caused by more general trafficking defects. The analysis uses a papss nonsense mutant from an EMS screen - I appreciate the rigorous approach the authors took to analyze transheterozygotes (as well as homozygotes) plus rescued animals in order to rule out effects of linked mutations. Importantly, the rescue experiments also demonstrated that sulfation enzymatic activity is important.The 2nd part of the paper focuses on the SG aECM, showing that Dpy and Pio ZP protein fusions localize abnormally in papss mutants and that these ZP mutants (and Np protease mutants) have similar SG lumen shaping defects to the papss mutants. A key conclusion is that SG lumen defects correlate with loss of a Pio+Dpy-dependent filamentous structure in the lumen. These data suggest that ZP protein misregulation could explain this part of the papss phenotype.Overall, the text is very well written and clear. Figures are clearly labeled. The methods involve rigorous genetic approaches, microscopy, and quantifications/statistics and are documented appropriately. The findings are convincing.Significance:This study will be of interest to researchers studying developmental morphogenesis in general and specifically tube biology or the aECM. It should be particularly of interest to those studying sulfation or ZP proteins (which are broadly present in aECMs across organisms, including humans).This study adds to the literature demonstrating the importance of luminal matrix in shaping tubular organs and greatly advances understanding of the luminal matrix in the Drosophila salivary gland, an important model of tubular organ development and one that has key matrix differences (such as no chitin) compared to other highly studied Drosophila tubes like the trachea.The detailed description of the defects resulting from papss loss suggests that there are multiple different sulfated targets, with a subset specifically relevant to aECM biology. A limitation is that specific sulfated substrates are not identified here (e.g. are these the ZP proteins themselves or other matrix glycoproteins or lipids?); therefore, it's not clear how direct or indirect the effects of papss are on ZP proteins. However, this is clearly a direction for future work and does not detract from the excellent beginning made here.Comments on revised version:Overall, I am pleased with the authors' revisions in response to my original comments and those of the other reviewers
**Reviewer #2 (Public review):**
SummaryThis study provides new insights into organ morphogenesis using the Drosophila salivary gland (SG) as a model. The authors identify a requirement for sulfation in regulating lumen expansion, which correlates with several effects at the cellular level, including regulation of intracellular trafficking and the organization of Golgi, the aECM and the apical membrane. In addition, the authors show that the ZP proteins Dumpy (Dpy) and Pio form an aECM regulating lumen expansion. Previous reports already pointed to a role for Papss in sulfation in SG and the presence of Dpy and Pio in the SG. Now this work extends these previous analyses and provides more detailed descriptions that may be relevant to the fields of morphogenesis and cell biology (with particular focus on ECM research and tubulogenesis). This study nicely presents valuable information regarding the requirements of sulfation and the aECM in SG development.Strengths-The results supporting a role for sulfation in SG development are strong. In addition, the results supporting the involvement of Dpy and Pio in the aECM of the SG, their role in lumen expansion, and their interactions, are also strong.-The authors have made an excellent job in revising and clarifying the many different issues raised by the reviewers, particularly with the addition of new experiments and quantifications. I consider that the manuscript has improved considerably.-The authors generated a catalytically inactive Papss enzyme, which is not able to rescue the defects in Papss mutants, in contrast to wild type Papss. This result clearly indicates that the sulfation activity of Papss is required for SG development.Weaknesses-The main concern is the lack of clear connection between sulfation and the phenotypes observed at the cellular level, and, importantly, the lack of connection between sulfation and the Pio-Dpy matrix. Indeed, the mechanism/s by which sulfation affects lumen expansion are not elucidated and no targets of this modification are identified or investigated. A direct (or instructive) role for sulfation in aECM organization is not clearly supported by the results, and the connection between sulfation and Pio/Dpy roles seems correlative rather than causative. As it is presented, the mechanisms by which sulfation regulates SG lumen expansion remains elusive in this study.-In my opinion the authors overestimate their findings with several conclusions, as exemplified in the abstract:"In the absence of Papss, Pio is gradually lost in the aECM, while the Dpy-positive aECM structure is condensed and dissociates from the apical membrane, leading to a thin lumen. Mutations in dpy or pio, or in Notopleural, which encodes a matriptase that cleaves Pio to form the luminal Pio pool, result in a SG lumen with alternating bulges and constrictions, with the loss of pio leading to the loss of Dpy in the lumen. Our findings underscore the essential role of sulfation in organizing the aECM during tubular organ formation and highlight the mechanical support provided by ZP domain proteins in maintaining luminal diameter."The findings leading to conclude that sulfation organizes the aECM and that the absence of Papss leads to a thin lumen due to defects in Dpy/Pio are not strong. The authors certainly show that Papss is required for proper Pio and Dpy accumulation. They also show that Pio is required for Dpy accumulation, and that Pio and Dpy form an aECM required for lumen expansion. However, the absence of Pio and Dpy do not fully recapitulate Papss mutant defects (thin lumen). I wonder whether other hypothesis and models could account for the observed results. For instance, a role for Papss affecting secretion, in which case sulfation would have an indirect role in aECM organization. This study does not address the mechanical properties of Dpy in normal and mutant salivary glands.-Minor issues relate to the genotype/phenotype analysis. It is surprising that the authors detect only mild effects on sulfation in Papss mutants using an anti-sulfoTyr antibody, as Papss is the only Papss synthathase. Generating germ line clones (which is a feasible experiment) would have helped to prove that this minor effect is due to the contribution of maternal product. The loss of function allele used in this study seems problematic, as it produces effects in heterozygous conditions difficult to interpret. Cleaning the chromosome or using an alternative loss of function condition (another allele, RNAi, etc...) would have helped to present a more reliable explanation.
**Recommendations for the authors:**

**Reviewer #1 (Recommendations for the authors):**
Overall, I am pleased with the authors' revisions in response to my original comments and those of the other reviewers. The addition of the sulfation(-) mutant to Fig. 1 is particularly nice. I have just a few additional suggestions for text changes to improve clarity/precision.(1) The current title of this manuscript is quite broad, making it sound like a review article. I recommend adding sulfation and salivary gland to the title to convey the main points more clearly. e.g. Sulfation affects apical extracellular matrix organization during development of the Drosophila salivary gland tube.

Thank you for the suggestion. We agree and have changed the title of the paper as suggested.

(2) Figure 1B shows very striking enrichment of papss expression in the salivary gland compared to other tubes like the trachea that also contain Pio and Dpy. To me, this implies that the key substrate(s) of Papss are likely to be unique, or at least more highly enriched, in the salivary gland aECM compared to the tracheal aECM (e.g. probably not Pio or Dpy themselves). I suggest that the authors address the implications of this apparent SG specificity in the discussion (paragraph beginning on p. 21, line 559).

Yes, we agree that there may be other key substrates of Papss in the SG, such as mucins, which play an important role in organizing the aECM and expanding the lumen. We have included a discussion.

(3) p. 15, lines 374-376 "The Pio protein is known to be cleaved, at one cleavage site after the ZP domain by the furin protease and at another cleavage site within the ZP domain by the matriptase Notopleural (Np) (Drees et al., 2019; Drees et al., 2023; Figure 5B)." As far as I can see, the Drees papers show that Pio is cleaved somewhere in the vicinity of a consensus furin cleavage site, but do not actually establish that the cleavage happens at this exact site or is done by a furin protease (this is just an assumption). Please word more carefully, e.g. "at one cleavage site after the ZP domain, possibly by a furin protease".

Thank you for pointing this out. We have edited the text.

**Reviewer #2 (Recommendations for the authors):**
Throughout the paper, I find a bit confusing the description of the lumen phenotype and their interpretations.Papss mutants produce SG that are either "thin" or show "irregular lumen with bulges". Do the authors think that these are two different manifestations of the same effect? or do they think that there are different causes behind?

The thin lumen phenotype appears to occur when the Pio-Dpy matrix is significantly condensed. When this matrix is less condensed in one region of the lumen than in other regions, the lumen appears irregular with bulges.

Are the defects in Grasp65 mutants categorized as "irregular lumen with bulges" similar to those in Papss mutants? Why do these mutants don't show a "thin lumen" defect?

Grasp65 mutant phenotypes are milder than those of Papss mutants. Multiple mutations in several Golgi components that more significantly disrupt Golgi structures and function may cause more severe defects in lumen expansion and shape.

How the defects described for Pio ("multiple constrictions with a slight expansion between constrictions") and Dpy mutants ("lumen with multiple bulges and constrictions") relate to the "irregular lumen with bulges" in Papss mutants?

pio and dpy mutants show more stereotypical phenotypes, while Papss mutants exhibit more irregular and random phenotypes. The irregular lumen phenotypes in Papss mutants are associated with a condensed Pio-Dpy matrix.